# When Random Saliency Looks Trained:
# Architectural Center Bias in CNN Interpretability

Keying Kuang [1]   Iain Carmichael [2]   Elizabeth Purdom [3][4]

## Abstract

Saliency maps are widely used to interpret image classification models and build trust in their predictions; however, their reliability remains a central concern, as randomized networks can produce saliency maps that closely resemble those of trained models. We identify a previously underappreciated architectural contributor to this phenomenon: a *center-focused saliency bias* induced by common convolutional design choices. Through controlled ablations, we show that architectural components such as zero padding and receptive field growth induce a center-focused saliency prior that is already present at randomly initialized CNNs and under randomized inputs. In contrast, this behavior is largely absent in non-convolutional architectures such as Vision Transformers (ViTs) and multilayer perceptrons (MLPs). To investigate the interaction between architectural priors and learning, we introduce a corner-shift benchmark and a Center-Shift Index that quantify how saliency redistributes under object relocation. We show that training can partially shift saliency toward object regions, while randomized models remain dominated by architectural center bias, providing one mechanism by which trained-random similarity can be inflated and clarifying how architectural priors can confound standard saliency evaluations.

[1]Division of Biostatistics, School of Public Health, University of California, Berkeley, CA, USA [2]School of Data Science and Society, Department of Pathology and Laboratory Medicine, University of North Carolina, Chapel Hill, NC, USA [3]Department of Statistics, Center for Computational Biology, University of California, Berkeley, CA, USA [4]Chan Zuckerberg Biohub, San Francisco, CA 94158. Correspondence to: Keying Kuang <keying_kuang@berkeley.edu>, Elizabeth Purdom <epurdom@stat.berkeley.edu>.

*Proceedings of the 43rd International Conference on Machine Learning*, Seoul, South Korea. PMLR 306, 2026. Copyright 2026 by the author(s).

## 1. Introduction

Saliency maps are among the most widely used tools for interpreting image classification models, aiming to highlight input regions that most strongly influence the predictions of a model, including classic gradient-based approaches such as Saliency, Integrated Gradients, SmoothGrad, Grad-CAM, and more recent attribution methods such as Expected Gradients (Simonyan et al., 2014; Sundararajan et al., 2017; Smilkov et al., 2017; Selvaraju et al., 2017; Erion et al., 2021). As these methods have gained popularity, a parallel line of research has emerged to evaluate whether saliency maps provide faithful and meaningful explanations of learned model behavior. However, assessing saliency maps in a principled way is still challenging.

One influential line of work introduced *sanity checks* (Adebayo et al., 2018) for saliency methods, showing that many widely used approaches produce visually similar saliency maps for trained and randomized CNNs. This surprising observation has frequently been interpreted as evidence that some saliency methods are insufficiently sensitive to model parameters and therefore unreliable as explanations. Subsequent work further examined the interpretation of such sanity checks, showing that they can depend strongly on the evaluation metric, randomization procedure, and explanation family (Binder et al., 2023; Yona & Greenfeld, 2021; Sixt et al., 2020; Kindermans et al., 2022; Bilodeau et al., 2024). Together, these studies suggest that trained–random similarity may arise from multiple interacting factors and that similarity-based evaluations should be interpreted carefully.

These observations raise a related question for saliency evaluation: to what extent can trained–random saliency similarity be influenced by properties of the model architecture, in addition to properties of the saliency method itself?

One plausible source is the spatial inductive bias of convolutional networks. CNNs are known to encode architectural biases, including preferences for local texture, translation sensitivity, and spatial aggregation effects arising from padding and receptive field growth (Zeiler & Fergus, 2014; Luo et al., 2016; Geirhos et al., 2019; Kayhan & van Gemert, 2020; Alsallakh et al., 2021). In particular, Alsallakh et al. (2021)

showed that zero padding can induce spatially nonuniform activation behavior near image boundaries. However, these architectural spatial effects have not been studied in the context of saliency-map evaluation or trained–random saliency similarity.

Our work connects these lines of research by showing that architecture-induced spatial bias can produce structured saliency patterns at initialization and substantially influence similarity-based saliency evaluations.

We provide evidence that architectural properties play a role in this similarity by combining empirical observation with targeted architectural interventions. We show that CNN architectures induce a *center-focused saliency prior* that is already present at initialization, before task-specific training, and whose observed strength can later be modulated by training. Randomly initialized CNNs consistently assign a higher saliency magnitude to central image regions than to peripheral regions, even when inputs are pixel-shuffled and contain no semantic structure. This phenomenon is architecture-dependent: it is pronounced in CNNs but largely absent in ViTs and MLP-based models (see Fig. 16 for an example).

We further demonstrate that this saliency center bias is a property of convolutional models, persisting across a wide range of CNN families rather than a single architecture.

This architectural prior also interacts with the spatial emphasis induced by training. In standard ImageNet training, where objects tend to be centered, training reshapes the saliency maps to reflect learned image content and object structure being emphasized but this emphasis still tends to be concentrated near the center. As a result, trained and random CNNs can have inflated saliency similarity under standard evaluation protocols, even when training has reshaped the saliency maps (shown in Fig. 6).

To disentangle architectural center bias from the training-induced spatial emphasis, we introduce a simple corner-shift training benchmark and propose the Center-Shift Index (CSI) to quantify how saliency responds to object location. This benchmark deliberately breaks the alignment between architectural center priors and the typical object placement in the training data, making it possible to assess how training reshapes saliency beyond the architecture center prior. Using this setup, we show that training can largely reweight saliency towards the object location, although residual center bias often remains in convolutional models. Together, our results identify architectural center bias as a concrete confounding factor in saliency evaluation and suggest that spatial stress tests are necessary to reliably isolate learned behavior.

Overall, this work emphasizes the importance of population-level evaluation in interpretability. By treating saliency maps as analyzable units with well-defined spatial proper-

ties, we are able to characterize systematic patterns across large datasets, model architectures, and experimental conditions, allowing us to compare beyond individual examples or specific use cases (see Figs. 12 and 13).

## 2. Qualitative Evidence of Architectural Center Bias

We begin by presenting a qualitative observation that motivates the remainder of the paper. Figure 1 shows saliency maps produced by a representative convolutional architecture (Inception-v4) under three conditions: a randomly initialized model evaluated on natural images, the same randomized model evaluated on pixel-shuffled images, and an ImageNet-trained model evaluated on natural images. Unless otherwise stated, we use vanilla gradients implemented in Captum (Kokhlikyan et al., 2020); results generalize to Integrated Gradients and SmoothGrad (Appendix D).

Across diverse inputs, randomly initialized CNNs consistently exhibit saliency maps that concentrate near the image center. Strikingly, this behavior persists even when the input images are pixel-shuffled, destroying all semantic structure and local edges. Prior work suggested saliency maps in randomized models largely reflect low-level edge or texture sensitivity (Adebayo et al., 2018). This observation that strong center-focused behavior still persists under pixel shuffling, a condition where such structures are destroyed, indicates that these explanations alone are insufficient, pointing to an additional architecture-induced spatial prior.

Training changes the spatial distribution of saliency by shifting emphasis toward object regions. However, even in trained models, a residual preference for central pixels remains visible. These observations indicate that center-focused saliency is not just a consequence of learning or data semantics but already present at initialization.

The following sections formalize this notion, identify its architectural origins, and explore its consequences for saliency evaluation.

## 3. Architectural Center Bias: Definition and Mechanism

These examples in the previous section provide intuitive evidence of center-focused saliency. To assess how systematic this behavior is across images, architectures, and saliency methods, we adopt a population-level perspective. Specifically, we treat each saliency map as a unit of analysis and compute summary statistics that capture different aspects of saliency structure which are then aggregated across inputs for comparison, as introduced in the following subsections.

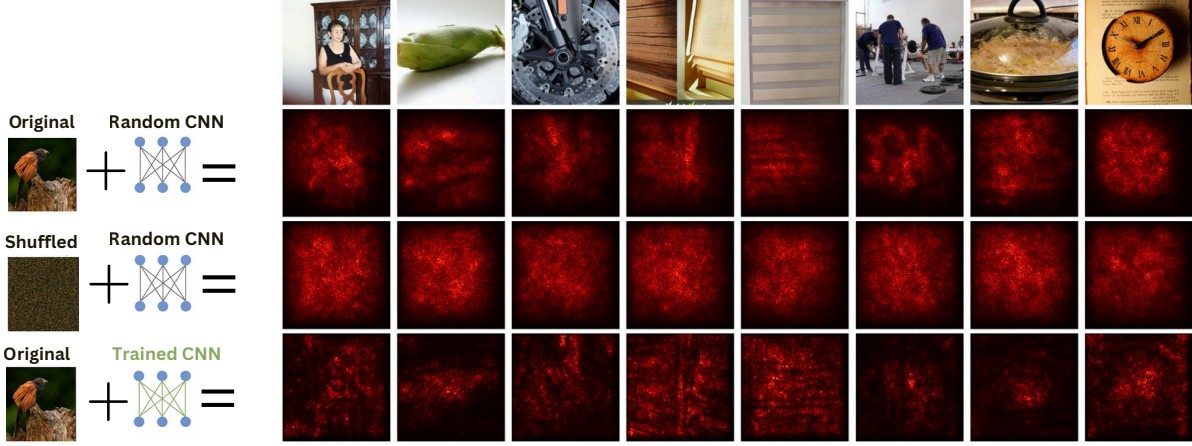

*Figure 1.* **Architectural center bias in CNN saliency maps.** Saliency maps (red) from an Inception-v4 model under three conditions: *(top)* randomly initialized CNN evaluated on original images, *(middle)* randomly initialized CNN evaluated on pixel-shuffled images, and *(bottom)* ImageNet-trained CNN evaluated on original images. Random CNNs show center-focused saliency regardless of image structure, including when object structure and edges are destroyed by pixel shuffling, indicating an architecture-induced spatial prior. Training shapes saliency to align with learned object structure, while remaining overall center-focused.

## 3.1. Measuring Architectural Center Bias

Let $A(x) \in \mathbb{R}^{H \times W}$ denote the saliency map produced by a model for an input image $x$. We work with the absolute saliency map normalized to a probability distribution over spatial locations:

$$\tilde{A}_{ij}(x) = \frac{|A_{ij}(x)|}{\sum_{u,v} |A_{uv}(x)|}, \qquad \sum_{i,j} \tilde{A}_{ij}(x) = 1.$$

$\tilde{A}_{ij}(x)$ represents the fraction of total saliency mass assigned to pixel $(i,j)$. This normalization factors out overall saliency magnitude, allowing center-focused behavior to be analyzed purely in terms of spatial allocation. Similar normalizations are standard in saliency analysis when the goal is to compare spatial concentration or regional importance rather than absolute scale (Fong & Vedaldi, 2017; Adebayo et al., 2018).

We quantify center bias using three complementary empirical metrics, each capturing a slightly different aspect of center-focused saliency behavior.

**Center–Bias AUC (CB-AUC).** We define a central square region $\mathcal{C} \subset \{1, \ldots, H\} \times \{1, \ldots, W\}$ as a square occupying one-half of the image width and height, and a peripheral region $\mathcal{K}$ as the union of four corner squares, each occupying one-quarter of the image width and height (Figure 3(a)). The Center–Bias AUC measures the extent to which saliency values in the center tend to exceed those in the periphery. Given a normalized saliency map $\tilde{A}_{ij}(x)$, it is defined as the empirical proportion of center–corner pixel pairs for which the center saliency is larger:

$$\text{CB-AUC}(x) = \frac{1}{|\mathcal{C}| \, |\mathcal{K}|} \sum_{(i,j) \in \mathcal{C}} \sum_{(u,v) \in \mathcal{K}} \mathbf{1} \Big[ \tilde{A}_{ij}(x) > \tilde{A}_{uv}(x) \Big].$$

This score lies in $[0, 1]$, with Center-Bias $\text{AUC}(x) = 0.5$ indicating no systematic spatial preference between the center and the periphery.

**Alternative metrics.** Formal definitions and implementation details for all metrics are provided in Appendix I. We use Center–Bias AUC as the primary measure in the main text, as it is bounded between 0 and 1 with a clear no-bias reference at 0.5, and is flexible with respect to region definitions. We further verify that architecture-level center-bias findings are stable across a range of center/corner region sizes; see Appendix H.1.

We additionally consider two alternative summaries in the appendix: a median difference between center and peripheral saliency values, and a radial cumulative AUC (RC-AUC) that measures how saliency mass accumulates with distance from the image center. These metrics provide complementary perspectives on center-focused behavior and we report these two metrics in Appendix D to assess robustness and provide a broader characterization of spatial saliency patterns across model families.

We say that a model family exhibits *architectural center bias* if these metrics consistently indicate excess saliency concentration near the image center under random initialization or on inputs lacking semantic structure, relative to a spatially uniform baseline.

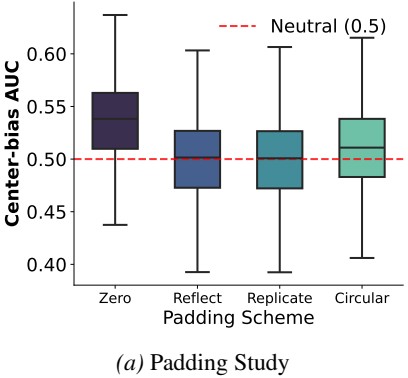

*(a)* Padding Study

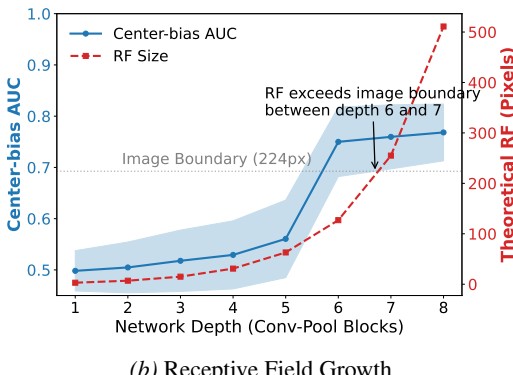

*(b)* Receptive Field Growth

*Figure 2.* **Mechanistic Origins of Center Bias.** We analyze the drivers of center-focused saliency using custom-designed, controlled CNN architectures that allows for systematic manipulation of architectural choices. **(a)**: Comparison across a standard 3-layer CNN shows zero-padding induces higher structural bias than alternative padding schemes. **(b)**: Using a depth-variable CNN to scale effective receptive field growth, reaching a saturation point when the RF exceeds the $224 \times 224$ image boundary; shaded regions indicate $\pm$ one standard deviation across random initializations.

### 3.2. Origin of Architectural Center Bias

The definition above treats center bias as a statistical property of a saliency map. We now explain why standard convolutional architectures induce such a bias even *before* training. The key observation is that the combination of zero padding and hierarchical aggregation produces spatially nonuniform activation and gradient statistics, systematically favoring central pixels over peripheral ones. Intuitively, zero padding reduces the number of valid convolutional contributions near image boundaries, leading to smaller pre-activation variance at peripheral locations. This variance imbalance propagates to gradient-based saliency through backpropagation under standard nonlinearities.

**Padding-induced variance imbalance.** Consider a single convolutional layer with kernel size $k \times k$, zero padding, and random weights $w_{uv} \sim \mathcal{N}(0, \sigma_w^2)$. The pre-activation at spatial location $(i, j)$ is $z_{ij} = \sum_{u,v} w_{uv}\, x_{i+u,\, j+v}$, where input values outside the image domain are zero due to padding. Conditioned on a fixed input $x$, the variance of $z_{ij}$ over random weights is

$$\mathrm{Var}(z_{ij}) = \sigma_w^2 \sum_{(u,v)\ \text{valid}} x_{i+u,\, j+v}^2, \qquad (1)$$

where "valid" kernel offsets $(u, v)$ are those for which the indices $(i + u,\, j + v)$ lie within the image boundary.

Interior pixels have $k^2$ valid kernel positions, whereas boundary pixels receive fewer contributions due to padding. Provided that input magnitudes do not systematically increase near image boundaries, this implies that the expected pre-activation variance is larger in the interior than near the image boundaries. A more detailed analysis linking padding, activation variance, and gradient magnitude is provided in Appendix A. **Implications:** As shown in Figure 2a,

replacing zero padding with alternative padding schemes helps "de-anchor" saliency from the image center. Since most modern CNN architectures (e.g. ResNet, Inception, VGG) use zero padding by default, this source of bias is broadly relevant in practice.

**From activation variance to gradient magnitude.** This forward-pass variance imbalance has direct implications for gradient-based saliency. At random initialization, backprop-agated input gradients at a given spatial location can be viewed as sums of many random contributions arising from different channels and computational paths. Standard concentration results for random vectors imply that the typical magnitude of such a gradient is governed by the variance of these contributions and their effective dimensionality. In particular, for a $D$-dimensional random vector with approximately independent, zero-mean sub-Gaussian entries, the squared $\ell_2$ norm concentrates around a value proportional to $D$ times the variance of its entries (Vershynin, 2018). Applying this principle to backpropagated gradients implies that spatial locations with larger pre-activation variance tend to exhibit larger typical gradient magnitudes. More details provided in Appendix A.

**Deepening induces Gaussian-like receptive fields.** As convolutional layers are stacked, the *effective receptive field* (RF) expands. Central pixels participate in more valid computational paths than peripheral pixels, amplifying the gradient magnitude difference introduced by padding. Empirically, we observe that center bias increases as the RF grows to span the image, but saturates once the RF exceeds the image dimensions and spatial specificity is lost (Figure 2b). Controlled ablations and implementation details supporting this behavior are provided in Appendix F.

Together, these effects explain why convolutional archi-

tectures exhibit systematically larger saliency magnitudes at central locations, even under random initialization and across degraded inputs such as pixel-shuffled images.

### 3.3. Illustrative Comparison Across Architecture Families

The mechanism described above makes a clear prediction: architectural center bias should arise in convolutional models that rely on spatial padding and hierachical receptive-field aggregation, but should be weak or absent in architectures without these design elements such as vision transformers and MLP-based models. As an illustrative test of this hypothesis, we compare saliency maps produced by a single representative model from each of these three architecture families: a convolutional neural network (Inception-v4), a vision transformer (ViT-B/16), and an MLP-based model (MLP-Mixer-B/16).

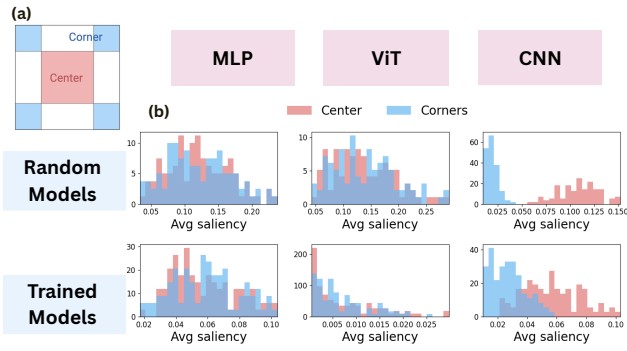

*Figure 3.* **Architectural center bias across model families. (a)** Center and corner masks used to aggregate saliency. **(b)** Distributions of median saliency values within center (red) and corner (blue) regions, computed across 100 images, for randomly initialized (top) and trained (bottom) models from three families: MLP-Mixer-B/16, ViT-B/16, and Inception-v4. CNNs exhibit center-focused saliency in both random and trained settings, whereas MLPs and ViTs show no systematic center bias.

Figure 3 shows the distribution of the median saliency mass in central and peripheral regions across 100 ImageNet images for each model. For Inception-v4, we observe a strong preference for central regions even under weight randomization, consistent with an architecture-induced spatial prior. In contrast, the ViT and MLP models exhibit substantially weaker center prior when randomized, indicating that the effect observed in Inception-v4 is not an inevitable property of saliency methods themselves.

Training reduces the magnitude of center concentration in Inception-v4, reflecting partial alignment between architectural priors and the learned semantics. However, because this comparison relies on a single representative model from each family, these observations are intended as illustrative rather than definitive.

## 4. Architectural Center Bias Across CNN Families

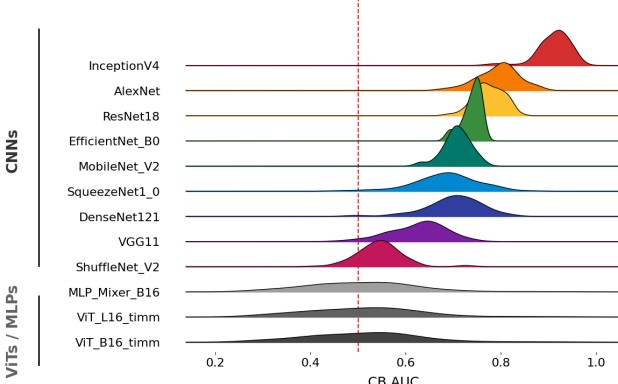

*Figure 4.* **Ridgeline distribution of Center-Bias AUC (CB-AUC) across architectures under random initialization.** Each ridge shows the distribution of `CB-AUC` values across images for a given model using a single random initialization. The red dashed line at 0.5 indicates the no-bias baseline. Convolutional architectures show consistently higher `CB-AUC` values, indicating strong center bias, whereas ViTs and MLPs (gray) show substantially weaker or absent center bias. Results are shown for vanilla gradients; corresponding results for SmoothGrad and Integrated Gradients, as well as additional center-bias metrics, are provided in Fig. 12, demonstrating that this pattern is consistent across saliency methods and estimators.

The illustrative comparison in the previous section raises a natural question: is the observed center-focused saliency behavior specific to a small number of canonical CNN models, or does it reflect a broader property of convolutional architectures?

To address this question, we evaluate center bias across a diverse collection of commonly used CNN architectures under random initialization, spanning shallow and deep models with varying receptive field sizes, depths, and architectural design choices. For each architecture, we compute center-bias scores over a shared set of 100 images using the estimators introduced in Section 3.1. In the main text, we report results using a single representative metric, `CB-AUC`, which measures the probability that saliency magnitude at a central location exceeds that at a peripheral (corner) location. This metric is bounded between 0 and 1, with a value of 0.5 corresponding to no systematic center preference.

Figure 4 summarizes the distribution of `CB-AUC` values across architectures using a single random initialization per model. The ridgeline distributions reflect variability across images rather than across random seeds; robustness across initialization seeds is provided in Appendix H.4. Despite substantial architectural diversity, most CNNs show a clear shift toward values greater than 0.5, indicating a consistent center-focused saliency preference. Architectures such as Inception-v4, AlexNet and ResNet18 display a particularly

strong center bias, while other CNNs—including ResNet, EfficientNet, MobileNet, DenseNet, and ShuffleNet variants show more moderate but still positive bias. In contrast, ViTs and MLP center around the no-bias baseline of 0.5, consistent with the absence of a systematic architectural center prior. While Figure 4 reports results using CB-AUC and vanilla gradients, we observe qualitatively consistent trends in multiple center-bias metrics, as well as in standard saliency methods including SmoothGrad and Integrated Gradients (Figure 12). These architecture-level rankings are also robust to the precise crop size used to define the center and corner regions (Appendix H.1). In particular, the relative ordering of architectures is preserved: ViTs and MLPs exhibit little to no center bias, whereas Inception-v4 and AlexNet consistently display the strongest bias.

These results reinforce the interpretation of center bias as an architectural prior inherent to convolutional networks rather than an artifact of a particular model or training configuration. While the magnitude of the bias varies across architectures, likely reflecting differences in depth, padding strategy, and receptive-field aggregation, the direction of the effect is remarkably stable across CNN designs. Full definitions and additional analyses are provided in Appendix I.

These findings demonstrate that architectural center bias is both *architecture-specific* (present in CNNs but not ViTs or MLPs) and *consistent across convolutional architectures*.

## 5. Why Architectural Center Bias Matters

The analyses above establish that center-focused saliency can arise from convolutional architecture prior to task-specific training and persists across a wide range of CNN designs, before task-specific training. We now examine why this architectural prior matters for interpretability evaluation and how it interacts with model training.

In practice, saliency maps are often inspected on individual examples to visualize the spatial distribution of gradient-based importance over the input. Both qualitative inspection and quantitative evaluation often rely on spatial patterns in saliency maps, either directly or through similarity-based metrics. If architectural center bias systematically influences saliency structure, then it may confound such evaluations by inducing apparent agreement that is unrelated to learned model behavior.

The practical impact of architectural center bias depends on its interaction with the spatial statistics of evaluation data. In standard ImageNet-style evaluation settings, objects are strongly concentrated near the image center at the dataset level (Appendix H.3). As a result, architectural center-focused saliency priors can align with the spatial distribution of objects in the data, inflating trained–random saliency similarity even when training has meaningfully

reshaped saliency maps.

To study this interaction directly, we design a controlled corner-shift benchmark that explicitly breaks the alignment between architectural center priors and dataset-level object statistics while preserving object identity and background structure (Fig. 6(a)(b)). Specifically, we construct two complementary training and evaluation conditions: (*i*) a **center-shrunk** condition, in which objects are uniformly downscaled by a fixed factor (0.5) and placed at the image center, and (*ii*) a **corner-shrunk** condition, in which the same downscaled objects are placed in the top-left corner of the image. The remaining image region is filled using pixel shuffling, which preserves the global color and intensity distribution of the original image while removing spatial and semantic structure in the background (Fig. 15). This design choice avoids introducing artificial background statistics (e.g. constant or low-variance fills) and ensures that the primary difference between conditions is object location rather than background composition. The corner-shrunk transformation preserves object identity while deliberately breaking the alignment between architectural center priors and dataset-level object location. All models are trained on the center-shrunk and corner-shrunk ImageNet variants using a consistent fine-tuning protocol; full training details are provided in Appendix G.1.

### 5.1. Training Interacts with Architectural Center Bias

Using the center- and corner-shrunk training conditions, we examine how training interacts with an architectural center prior by quantifying how saliency mass redistributes when object location is placed in conflict with that prior. To this end, we introduce the *Center-Shift Index* (CSI), a scalar measure of how far saliency mass shifts away from the image center toward the object location.

Let $\tilde{A}(x) \in \mathbb{R}^{H \times W}$ denote the normalized saliency map defined in Section 3.1. Let $(x_{\text{com}}, y_{\text{com}})$ be the center of mass of $\tilde{A}(x)$, and let $c = (c_x, c_y) = \left(\frac{H-1}{2}, \frac{W-1}{2}\right)$ denote the image center. For the top-left corner benchmark setting, we define the signed Center-Shift Index (CSI) as

$$\text{CSI}(x) = \frac{(c_x - x_{\text{com}}) + (c_y - y_{\text{com}})}{c_x + c_y}.$$

Here image coordinates follow the standard convention with the origin at the top-left corner. By construction, $\text{CSI} \approx 0$ indicates saliency concentrated near the image center, while larger values indicate increasing displacement toward the target top-left corner. Negative values indicate displacement in the opposite direction. For the default corner-shrunk benchmark setting with square inputs ($c_x = c_y$), perfect alignment of saliency with the object location corresponds to an ideal $\text{CSI} \approx 0.5$.

Figure 5 illustrates the CSI geometry and reports empirical

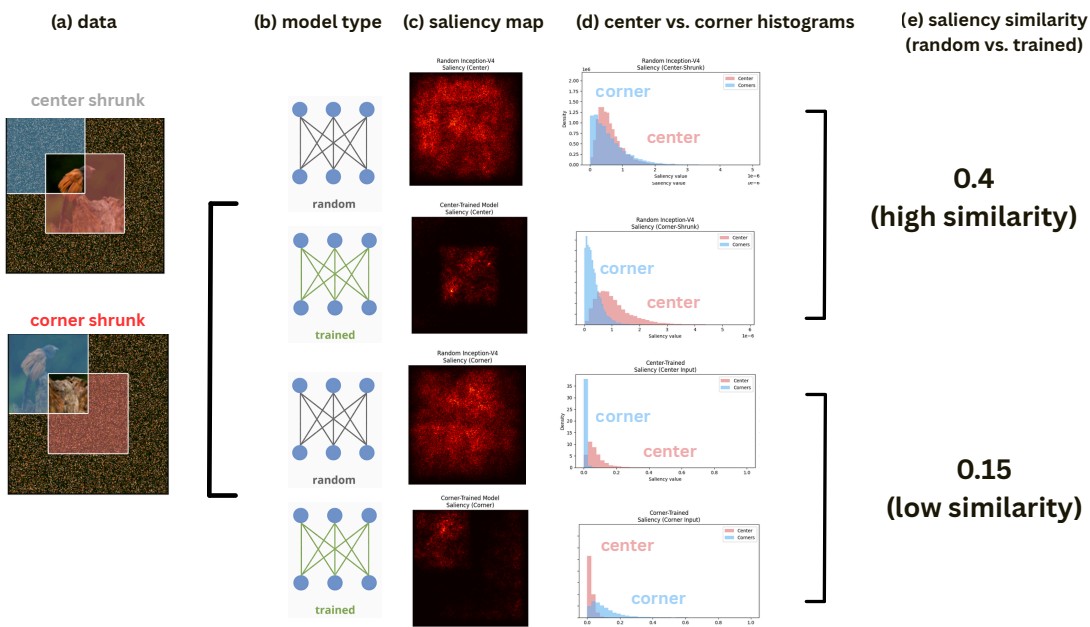

Figure 5. **Training shifts saliency toward object location, quantified by the Center-Shift Index (CSI).** *Left:* Conceptual illustration of CSI. CSI measures the displacement of saliency mass relative to the image center. For center-shrunk inputs, saliency aligned with the object yields an expected CSI $\approx 0$. For corner-shrunk inputs, perfect alignment with the object location corresponds to an ideal CSI $\approx 0.5$. Intermediate values indicate partial shifts, reflecting residual architectural center bias. *Right:* Empirical CSI distributions across architectures and saliency methods, showing that training substantially shifts saliency toward object location, while residual center bias remains.

Figure 6. **Object position affects trained–random saliency similarity through architectural center bias.** (a) Center-shrunk and corner-shrunk inputs constructed by spatially relocating the same object. (b) Trained or randomly initialized Inception-V4 models. (c) Resulting saliency maps for each model. (d) Distributions of saliency mass within benchmark-specific center and corner masks (shown in (a)), illustrating that random models remain center-focused regardless of object location, whereas trained models shift saliency toward the object. (e) Corresponding trained–random saliency *map* similarity scores, which are high when objects are centered ($\rho \approx 0.40$) but decrease when objects are moved to the corner ($\rho \approx 0.15$).

CSI distributions across architectures and saliency methods. Training on corner-shrunk images shifts saliency toward the object location for all architectures, indicating that learned representations can partially override the architectural center prior. Similar qualitative trends are also observed under adversarially robust fine-tuning (Appendix H.5). However,

CSI values remain below the expected reference of 0.5, indicating that saliency mass does not fully relocate to the displaced object location and that architectural center bias may continue to shape spatial attribution patterns after training. Additional ablations show that this shift persists under alternative object scales and placements, indicating that the CSI trend is not tied to the specific top-left, 0.5-scale construction used in the main benchmark (Appendix H.2). Example saliency maps showing these center- and corner-shift conditions across architectures are shown in Appendix K (Figs. 17–18).

Classification accuracies reported in Appendix G.2 further indicate that models learn the task comparably under both center- and corner-shrunk conditions, suggesting that this effect is not due to inadequate task learning.

To complement CSI, Appendix C reports a benchmark-based analysis using the standard Center-Bias AUC metric under the same spatial intervention. While CSI quantifies how far saliency mass shifts with object displacement, the Center-Bias AUC comparison captures changes in relative center dominance. Both measures show consistent directional effects across architectures (Fig. 10).

## 5.2. Architectural Center Bias Inflates Trained–Random Saliency Similarity

We next examine how architectural center bias interacts with a common evaluation protocol (Adebayo et al., 2018) that quantifies similarity between saliency maps from trained and randomized models. Figure 6 illustrates this effect for a representative CNN architecture, Inception-v4.

For models with relatively high center bias (e.g., Inception-v4 and AlexNet), when objects are centered, the architectural center bias aligns with training data, producing saliency maps from trained and random models that are spatially similar in the sense that both concentrate mass near the image center, despite differences in the detailed saliency structure. As a result, trained–random similarity scores are relatively high under standard centered evaluation settings for CNN models, where objects are typically located near the center. When objects are moved to the corner, this alignment is disrupted: trained models redistribute saliency toward the object location (e.g., the upper-left corner), while random models retain center-focused saliency pattern, leading to a reduction in similarity scores.

To make sure the effect is robust across similarity definitions, we evaluate a range of saliency similarity metrics spanning three broad categories: pixelwise (e.g., Spearman, Spearman-Abs, and Pearson Corr), structure-based (e.g., SSIM and Pearson HOG), and location-aware metrics that explicitly account for global spatial alignment. The complete comparison across eight metrics and architectures is

shown in Figure 11 (Appendix J), including metrics commonly used in prior saliency sanity-check studies, as well as several spatially aware alternatives introduced for this benchmark; formal definitions and implementation details for all metrics are provided in the appendix.

In the main text, we focus on two representative similarity metrics, structure-based Pearson HOG and location-aware Blur-Pearson shown in Figure 7.

Blur-Pearson emphasizes coarse spatial alignment by measuring similarity between saliency maps after spatial smoothing, thereby reducing sensitivity to local edge structure and highlighting alignment of broader saliency regions. Pearson-HOG measures Pearson correlation between Histogram of Oriented Gradients (HOG) extracted from saliency maps, capturing similarity in local edge orientation and structural patterns while being less sensitive to absolute spatial location.

Across architectures with strong architectural center bias (e.g., Inception-v4 and AlexNet), these metrics consistently report higher trained-random similarity under center-aligned inputs than under corner-shifted inputs. Architectures with weaker or less dominant center priors tend to exhibit smaller or less consistent differences between center- and corner-shifted conditions. In these cases, similarity becomes more influenced by architecture-specific saliency structure rather than by global center alignment alone. For example, some models produce more diffuse or background-dominated saliency patterns under random initialization (Appendix Figs. 17–18)), which can reduce the center–corner gap or even produce negative correlation-based similarity scores despite the presence of object-focused saliency in trained models. As a result, when the shared global center prior is weak, trained–random similarity becomes more influenced by model-specific saliency structure (e.g., diffuse versus object-focused patterns), whereas stronger center-biased architectures are more consistently dominated by spatial center alignment.

In light of these results, architectural center bias can act as a confounding factor in similarity-based saliency evaluation. High trained–random similarity does not necessarily indicate insensitivity of saliency methods to learned parameters, but can arise when architectural spatial priors align with the spatial distribution of evaluation inputs. At the same time, the magnitude and even direction of similarity differences depend on both the underlying architecture and the choice of similarity metric, underscoring that such scores must be interpreted in context rather than as absolute indicators of explanation quality. Simple spatial stress tests, such as object relocation or center-shift benchmarks, provide a practical way to investigate the influence of architectural bias and contextualize similarity-based saliency evaluations.

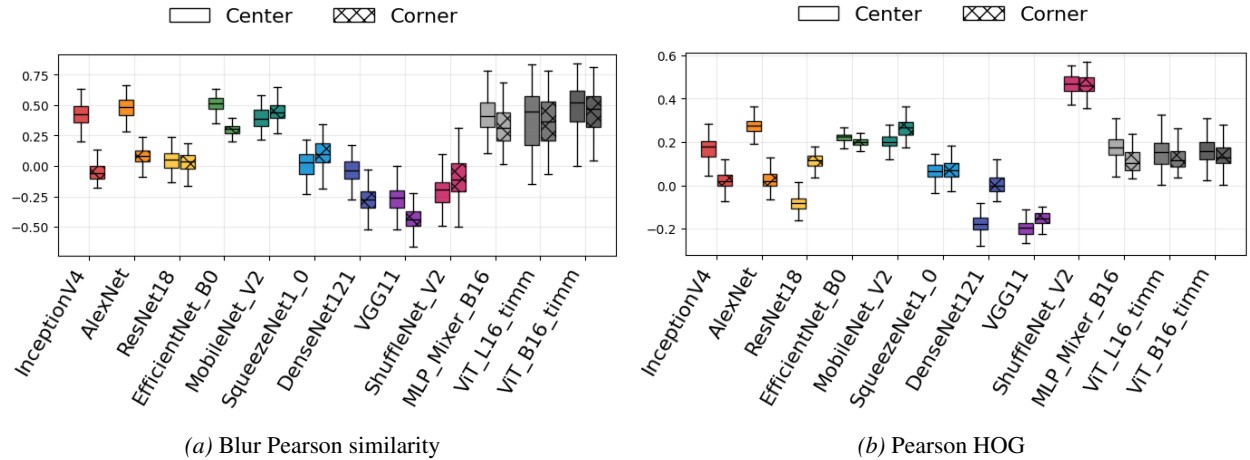

*(a)* Blur Pearson similarity        *(b)* Pearson HOG

*Figure 7.* **Trained–random saliency similarity under center- and corner-shifted inputs.**

## 6. Conclusion.

This paper provides an architectural perspective on why randomized convolutional networks can produce gradient-based saliency maps that are highly similar to those of trained models. Past work has largely attributed trained–random saliency similarity to properties of the saliency methods themselves (e.g. edge-detector effects in (Adebayo et al., 2018)); we show that architectural choices such as zero padding and hierarchical aggregation induce systematic spatial structure in gradients at initialization, which can confound similarity-based evaluations. While architectural center bias is not the only factor influencing trained–random similarity, our results suggest that these systematic spatial biases are important and previously under-characterized factors in similarity-based saliency evaluations.

These architectural spatial priors, by introducing biases in the center of the image, interact with the spatial statistics of standard image datasets, where objects are also often concentrated near the image center. As a result, trained and random CNN saliency maps can exhibit inflated similarity under standard evaluation protocols even when training has meaningfully reshaped saliency structure.

To study this interaction, we introduced the corner-shift benchmark and the Center-Shift Index (CSI), which explicitly break the alignment between architectural center priors and dataset-level object statistics. These controlled spatial stress tests help disentangle architectural spatial effects from learned object-focused behavior and provide a practical diagnostic tool for evaluating similarity-based interpretability protocols.

Our analysis focuses primarily on image classification architectures and gradient-based saliency methods, and additional work is needed to understand how these effects extend to other modalities and explanation families. In addition, our focus is complementary to fidelity-based evaluation metrics (e.g. insertion/deletion or prediction-faithfulness measures), which assess whether saliency maps reflect model behavior; instead, we study how architectural spatial priors can systematically influence saliency structure and confound similarity-based evaluation protocols. More broadly, we hope this work encourages interpretability evaluations that account not only for the properties of saliency methods, but also for the architectural priors of the underlying models.

**Code Availability**    Code for reproducing the experiments and figures is available at: `https://github.com/keyingkuang/saliency-center-bias`.

## Acknowledgements

We thank Van Hovenga and Li Liu for their helpful discussions. E. P. is a Chan Zuckerberg Biohub investigator.

## Impact Statement

This paper presents work whose goal is to advance the field of machine learning, specifically in the evaluation of interpretability methods for image classification models. There are many potential societal consequences of improved model interpretability, including more reliable deployment of machine learning systems; however, we do not identify any immediate ethical concerns or negative societal impacts that require specific discussion here.

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

# A. Link between padding, activation variance, and gradient magnitude

This appendix provides a simplified mechanistic explanation to clarify the origin of the observed center bias in randomly initialized CNN models, rather than a full characterization of gradient distributions in deep networks.

## A.1. Padding-induced spatial variance in pre-activations

**Setup.** Let the input image be

$$x \in \mathbb{R}^{C \times H \times W},$$

with $C$ channels and spatial dimensions $H \times W$. Consider a single convolutional layer with kernel size $k \times k$, and define $r$ such that

$$k = 2r + 1, \qquad r = \lfloor k/2 \rfloor.$$

We assume odd kernel sizes (as is standard in CNNs) so that the convolution is centered and admits a symmetric receptive field of radius $r$.

Define the zero-padded input

$$\tilde{x} \in \mathbb{R}^{C \times (H+2r) \times (W+2r)}$$

by

$$\tilde{x}_{c,i,j} = \begin{cases} x_{c,i-r,j-r}, & \text{if } 1 \le i - r \le H, \ 1 \le j - r \le W, \\ 0, & \text{otherwise.} \end{cases}$$

Let the convolutional weights be

$$w_{c,u,v} \sim \mathcal{N}(0, \sigma_w^2), \quad c = 1, \dots, C, \ u, v \in \{-r, \dots, r\},$$

independent across all indices. Let $z_{i,j,d}$ denote the pre-activation at spatial location $(i, j)$ and output channel $d$. For notational simplicity, we first consider a single output channel and temporarily suppress the index $d$; the multi-channel case is then handled explicitly in Section A.3. We also ignore bias terms which only add spatially constant offsets and do not affect spatial variance comparisons. The pre-activation at spatial location $(i, j)$ is defined as

$$z_{i,j} = \sum_{c=1}^{C} \sum_{u=-r}^{r} \sum_{v=-r}^{r} w_{c,u,v} \, \tilde{x}_{c,i+u,j+v}. \tag{A.1}$$

Throughout this appendix, all expectations and variances are taken with respect to random network weights at initialization, conditioning on a fixed input image $x$ and fixed spatial locations $(i, j)$.

## A.2. Variance of pre-activations under random weights

Conditioned on a fixed input $x$, $z_{i,j}$ is a mean-zero random variable over the randomly initialized weights.

By independence of the weights,

$$\begin{aligned}
\mathrm{Var}_w(z_{i,j}) &= \sum_{c=1}^{C} \sum_{u=-r}^{r} \sum_{v=-r}^{r} \mathrm{Var}(w_{c,u,v}) \, \tilde{x}_{c,i+u,j+v}^2 \\
&= \sigma_w^2 \sum_{c=1}^{C} \sum_{u=-r}^{r} \sum_{v=-r}^{r} \tilde{x}_{c,i+u,j+v}^2.
\end{aligned} \tag{A.2}$$

Define the number of valid (non-padded) kernel positions at spatial location $(i, j)$ as

$$m(i,j) = \# \left\{ (u,v) \in [-r, r]^2 : 1 \le i + u - r \le H, \ 1 \le j + v - r \le W \right\}. \tag{A.3}$$

If $(i, j)$ satisfies $r + 1 \le i \le H + r$ and $r + 1 \le j \le W + r$, then $m(i, j) = k^2$. If $(i, j)$ lies near the boundary, then $m(i, j) < k^2$ due to zero padding. Equation (A.2) shows that the pre-activation variance depends on the amount of input

energy falling inside the valid convolutional window. Since boundary locations have fewer valid input terms, zero padding can reduce this variance near the image boundary unless boundary pixels have systematically larger squared magnitude than interior pixels.

To make this intuition explicit, suppose that the input has approximately spatially homogeneous second moments, so that

$$\frac{1}{C}\sum_{c=1}^{C} x_{c,p,q}^2 \approx \tau^2 \quad \text{uniformly over spatial locations } (p,q).$$

Under this approximation, spatial differences in $\mathrm{Var}_w(z_{i,j})$ are primarily governed by the number of valid non-padded kernel positions $m(i,j)$, giving

$$\mathrm{Var}_w(z_{i,j}) = \sigma_w^2 \sum_{c=1}^{C}\sum_{(u,v)\text{ valid}} x_{c,i+u-r,j+v-r}^2$$
$$\approx \sigma_w^2\, C\, \tau^2\, m(i,j). \tag{A.4}$$

Thus, under spatially homogeneous input energy, locations with fewer valid kernel positions are expected to have smaller pre-activation variance. This provides a simple mechanism by which zero padding can introduce spatially nonuniform activation statistics before any training occurs.

### A.3. From activation variance to typical activation magnitude

We now generalize from scalar pre-activations to the vector of pre-activations across output channels. For fixed $(i,j)$, the scalar pre-activation $z_{i,j,d}$ is Gaussian with variance given by (A.2), hence

$$\mathbb{E}_w[z_{i,j,d}^2] = \mathrm{Var}_w(z_{i,j,d}). \tag{A.5}$$

More generally, if we consider $D$ output channels and stack the corresponding pre-activations into a vector $z_{i,j} \in \mathbb{R}^D$, then

$$\mathbb{E}\|z_{i,j}\|_2^2 = \sum_{d=1}^{D} \mathrm{Var}(z_{i,j,d}). \tag{A.6}$$

Substituting Eq. (A.4) yields

$$\mathbb{E}\|z_{i,j}\|_2^2 \approx D\,\sigma_w^2\, C\, \tau^2\, m(i,j),$$

making explicit that spatial variation in the typical activation magnitude arises solely through the number of valid kernel positions $m(i,j)$.

Standard results on the concentration of high-dimensional random vectors (e.g., Vershynin (2018)) show that, when the dimension $D$ is moderate to large, the squared $\ell_2$ norm of a random vector with approximately independent sub-Gaussian components is closely related to the variance of its entries and the number of dimensions. $\|z_{i,j}\|_2^2$ is typically of the same order as its expectation $\mathbb{E}\|z_{i,j}\|_2^2$, with relatively small fluctuations. In modern CNNs, $D$ corresponds to the number of output channels and is commonly in the tens to hundreds, making this approximation appropriate in practice.

Therefore, larger pre-activation variance implies larger typical activation magnitude. At random initialization, both of the pre-activations and backpropagated gradients can be viewed as sums of many random contributions (products of weights and activations), and can be well-approximated as sub-Gaussian in random-network analyses.

### A.4. Gradient propagation and variance inheritance

Let $f(x)$ denote a scalar network output. For an input pixel $x_{c,p,q}$, the chain rule gives

$$\frac{\partial f}{\partial x_{c,p,q}} = \sum_{i,j} \frac{\partial f}{\partial z_{i,j}} \frac{\partial z_{i,j}}{\partial x_{c,p,q}}. \tag{A.7}$$

We define the receptive field $RF(i, j)$ as the set of input spatial locations $(p, q)$ that influence the pre-activation $z_{i,j}$. From (A.1),

$$\frac{\partial z_{i,j}}{\partial x_{c,p,q}} = \begin{cases} w_{c,u,v}, & \text{if } (p, q) = (i + u - r, j + v - r) \text{ and } 1 \leq p \leq H, \ 1 \leq q \leq W, \\ 0, & \text{otherwise.} \end{cases} \tag{A.8}$$

Hence, only spatial locations $(i, j)$ for which $(p, q)$ lies within the receptive field contribute to the gradient.

Under standard independence assumptions at random initialization (neglecting cross-covariance terms), the gradient variance can be approximated as

$$\text{Var}\left(\frac{\partial f}{\partial x_{c,p,q}}\right) \approx \sum_{i,j} \mathbb{E}\left[\left(\frac{\partial f}{\partial z_{i,j}}\right)^2\right] \text{Var}\left(\frac{\partial z_{i,j}}{\partial x_{c,p,q}}\right). \tag{A.9}$$

Equation A.9 follows by neglecting cross-covariance terms between gradient contributions from different spatial locations (standard in analyses of random weight networks). This approximation is exact when such contributions are independent and captures the leading-order scaling of the variance that keep the gradient variance consistent across layers.

Using (A.8) and $\text{Var}(w_{c,u,v}) = \sigma_w^2$,

$$\text{Var}\left(\frac{\partial z_{i,j}}{\partial x_{c,p,q}}\right) = \begin{cases} \sigma_w^2, & \text{if } (p, q) \in \text{RF}(i, j), \\ 0, & \text{otherwise.} \end{cases} \tag{A.10}$$

Moreover,

$$\frac{\partial f}{\partial z_{i,j}} = \frac{\partial f}{\partial h_{i,j}} \phi'(z_{i,j}), \tag{A.11}$$

so the distribution of the upstream gradient depends explicitly on the distribution of $z_{i,j}$. In particular, moments of $\phi'(z_{i,j})$ depend on the variance of $z_{i,j}$ for common nonlinearities.

Combining (A.9)–(A.11), spatial locations with larger pre-activation variance induce larger gradient variance. By the same variance–norm relationship used in (A.6), larger gradient variance implies larger typical gradient magnitude.

### A.5. Implication for saliency maps

Recall that the saliency map $A(x) \in \mathbb{R}^{H \times W}$ is defined by

$$A_{p,q}(x) = \left\| \nabla_{x_{:,p,q}} f(x) \right\|_2, \tag{A.12}$$

The analysis shows that spatial non-uniformity in pre-activation variance induced by zero padding propagate through the network and result in systematic spatial differences in the typical magnitude of $A_{p,q}(x)$, providing a plausible mechanistic explanation for center-biased gradient-based saliency maps in randomly initialized convolutional networks.

## B. Empirical Diagnostics for Activation Variance

Appendix A argues that zero padding and hierarchical aggregation can induce spatially nonuniform activation variance, which in turn contributes to center-focused gradient magnitude. This section provides empirical diagnostics for the activation-variance component of this mechanism. The goal is not to fully identify every backpropagation path in a deep network, but rather to test whether the spatial variance patterns predicted by the mechanism are observed in randomly initialized convolutional architectures.

We evaluate randomly initialized ConvNeXt-Tiny as an additional modern convolutional architecture. Although ConvNeXt-Tiny is not used as a primary model in the main benchmark, it provides a useful diagnostic because its depthwise convolutional stages retain local spatial aggregation while using a more recent CNN design. We hook the first depthwise convolution in each ConvNeXt stage, before subsequent normalization and pointwise mixing, and measure pre-activation variance across random initializations.

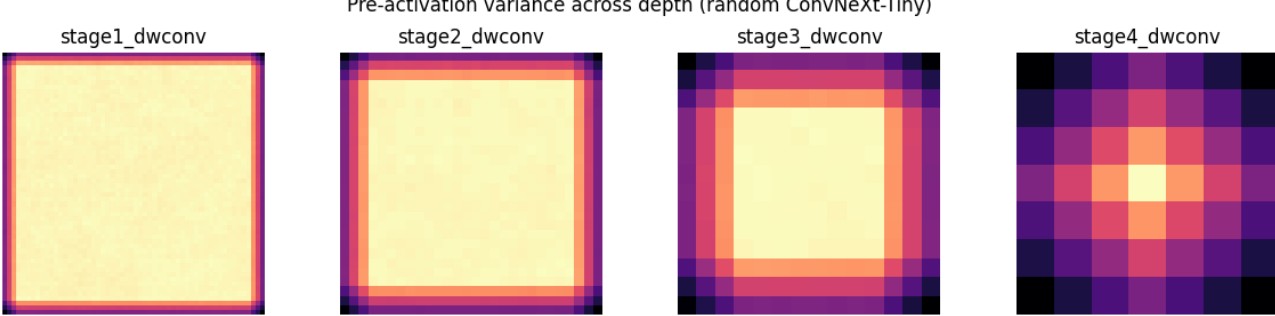

*Figure 8.* **Pre-activation variance maps across depth in random ConvNeXt-Tiny.** For each ConvNeXt stage, we compute pre-activation variance across random initializations, then average across channels and images. Central spatial locations exhibit consistently higher variance than boundary locations, while the spatial contrast between center and boundary regions becomes increasingly structured with depth.

For each fixed input image and spatial location, we compute variance across random seeds, average across channels, and then average across images. Specifically, for layer $\ell$, image $n$, seed $s$, channel $d$, and spatial location $(i, j)$, let $z^{(\ell)}_{s,n,d,i,j}$ denote the hooked activation. We estimate the variance over random initializations (seeds),

$$v^{(\ell)}_{n,d,i,j} = \mathrm{Var}\left(z^{(\ell)}_{s,n,d,i,j}\right),$$

then average this variance over channels and images:

$$V^{(\ell)}(i,j) = \frac{1}{N}\sum_{n=1}^{N}\frac{1}{D_\ell}\sum_{d=1}^{D_\ell} v^{(\ell)}_{n,d,i,j}.$$

This is an average-of-variances estimator, matching the conditioning in Appendix A, rather than the variance of an averaged activation.

Figure 8 visualizes the resulting spatial variance maps across ConvNeXt-Tiny stages. Even under random initialization, activation variance is consistently higher near the image center than near the boundary. Moreover, the spatial contrast between center and boundary regions becomes increasingly structured at deeper stages.

To summarize these effects spatially, Figure 9 shows radial variance profiles plotted in input-pixel units by accounting for the effective stride of each stage. Across all stages, activation variance decays smoothly from the image center toward the boundary, consistent with the spatial variance patterns predicted by Appendix A.

Overall, we observe three consistent patterns across stages: central locations exhibit higher activation variance than boundary locations; the spatial contrast between center and boundary regions becomes more structured with depth; and radial variance decays smoothly from the image center toward the periphery. These observations are consistent with the mechanism proposed in Appendix A: padding and receptive-field aggregation induce spatially nonuniform activation statistics, which provide a plausible source of the center-focused gradient patterns observed in randomly initialized CNNs.

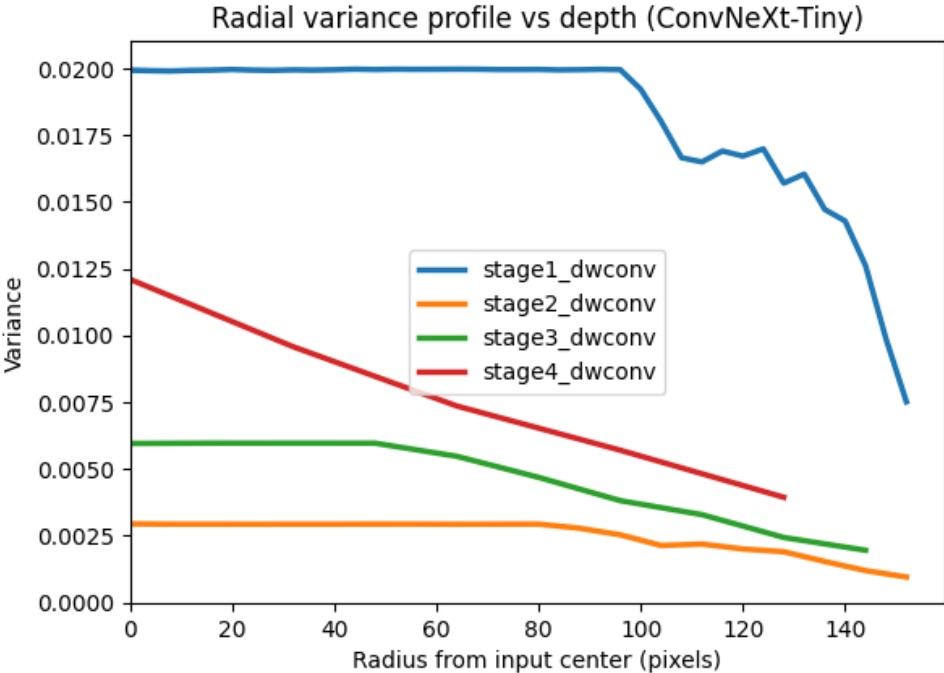

*Figure 9.* **Radial variance profiles across depth in random ConvNeXt-Tiny.** Radial profiles are plotted in input-pixel units by accounting for the effective stride of each stage. Across all stages, activation variance decays smoothly from the image center toward the boundary.

## C. Additional Analysis of Saliency Shift Metrics

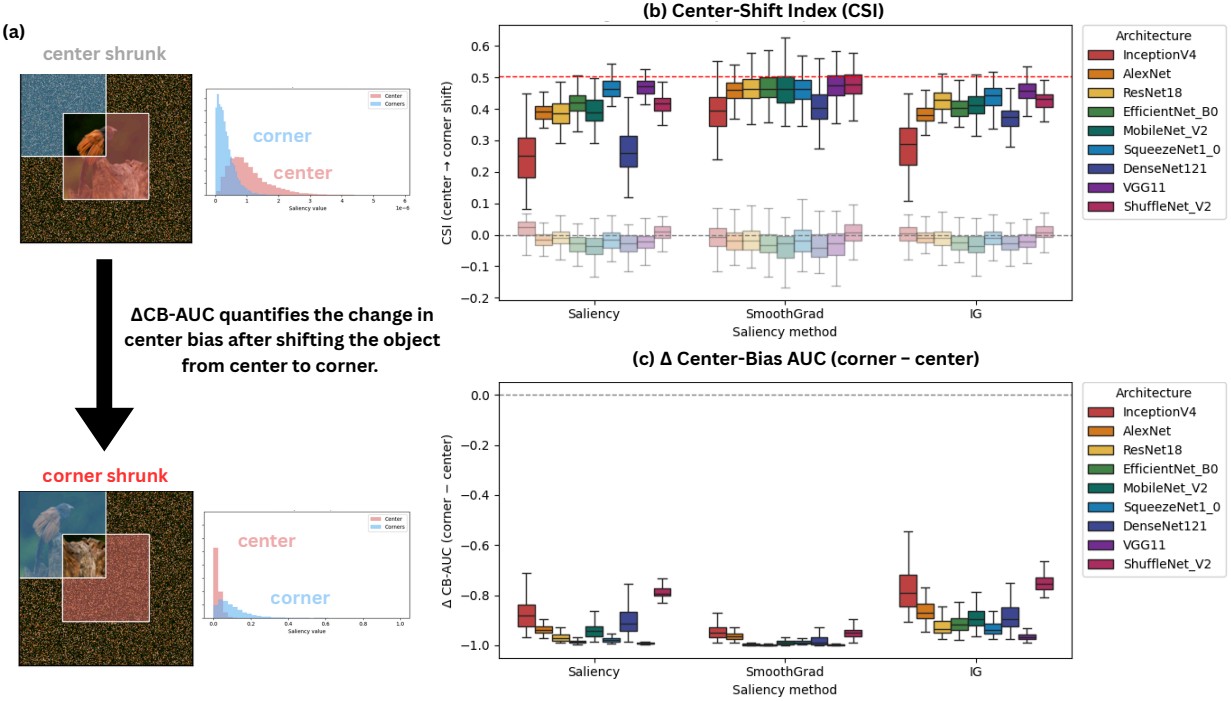

*Figure 10.* **Complementary measures of saliency shift under spatial displacement. (a)** Illustration of the center–corner benchmark and the definition of $\Delta$CB-AUC. Saliency maps are evaluated on inputs where the object is centered (top) or shifted to the corner (bottom), using fixed center and corner regions of equal area. $\Delta$CB-AUC measures the change in relative center dominance after shifting the object from the center to the corner. **(b)** Center-Shift Index (CSI) for trained CNNs under center-shrunk (transparent) and corner-shrunk (solid) inputs. CSI directly quantifies the fraction of saliency mass displaced toward the object location, with an ideal value of 0 when saliency is fully centered and 0.5 when fully concentrated in the corner. **(c)** Change in Center-Bias AUC ($\Delta$CB-AUC = corner – center) computed using the same benchmark regions. More negative values indicate a stronger reduction in center bias when the object is displaced.

We provide a complementary view of saliency shift using both the Center-Shift Index (CSI) and a benchmark-based Center-Bias AUC metric, where the center and corner benchmark regions are illustrated in Fig. 10 (a). CSI measures the absolute displacement of saliency mass under a center to corner intervention, and is therefore visualized by overlaying the baseline (center-shrunk, transparent) and shifted to corner (corner-shrunk, solid) distributions, with an interpretable scale where values near 0 indicate centered saliency and values near 0.5 indicate saliency fully concentrated at the corner. The Center-Bias AUC metric compares relative saliency strength between predefined center and corner regions for a saliency map, and does not have an absolute spatial reference. We therefore report the change in Center-Bias AUC between corner-shrunk and center-shrunk inputs ($\Delta$CB-AUC = corner - center), where more negative values indicate a stronger reduction in center focus after the shift. Although this benchmark relies on a coarse, fixed center–corner mask rather than an ideal spatial reference, architectures exhibiting larger CSI values generally show more negative $\Delta$CB-AUC, indicating broadly aligned directional trends across the two measures.

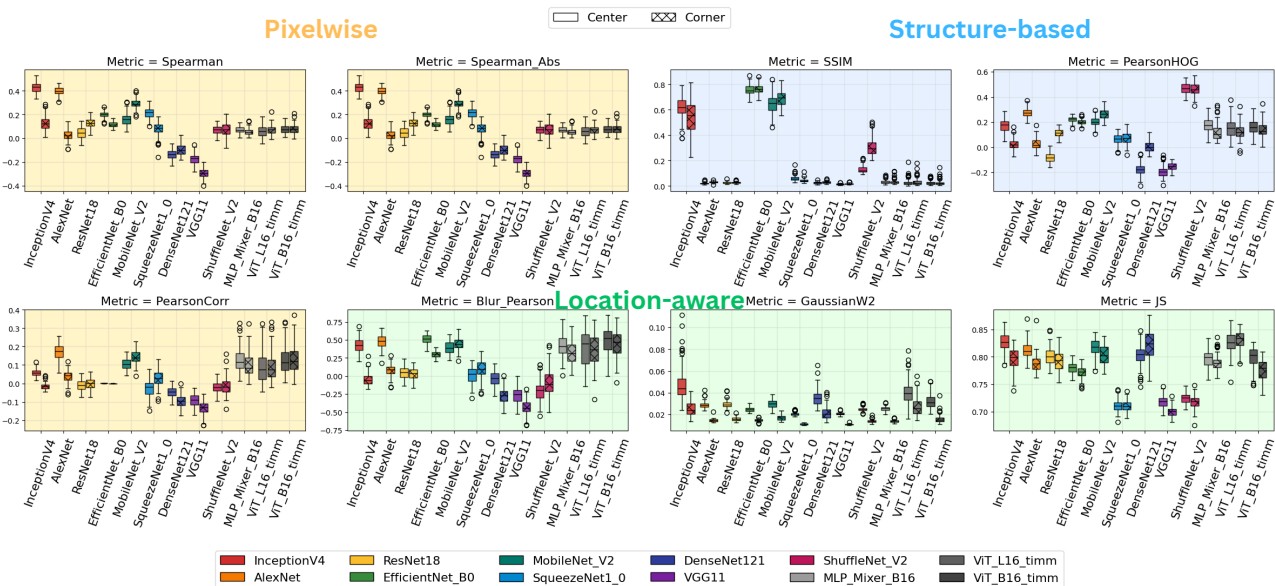

*Figure 11.* Comparison of eight saliency similarity metrics across architectures for center-trained vs. corner-trained models. Metrics fall into three categories: (1) **Pixelwise**: Spearman, Spearman-Abs, PearsonCorr; (2) **Local structure**: SSIM, PearsonHOG; (3) **Location-aware**: Blur-Pearson, GaussianW2, JS.

# D. Extended Results and Visualizations

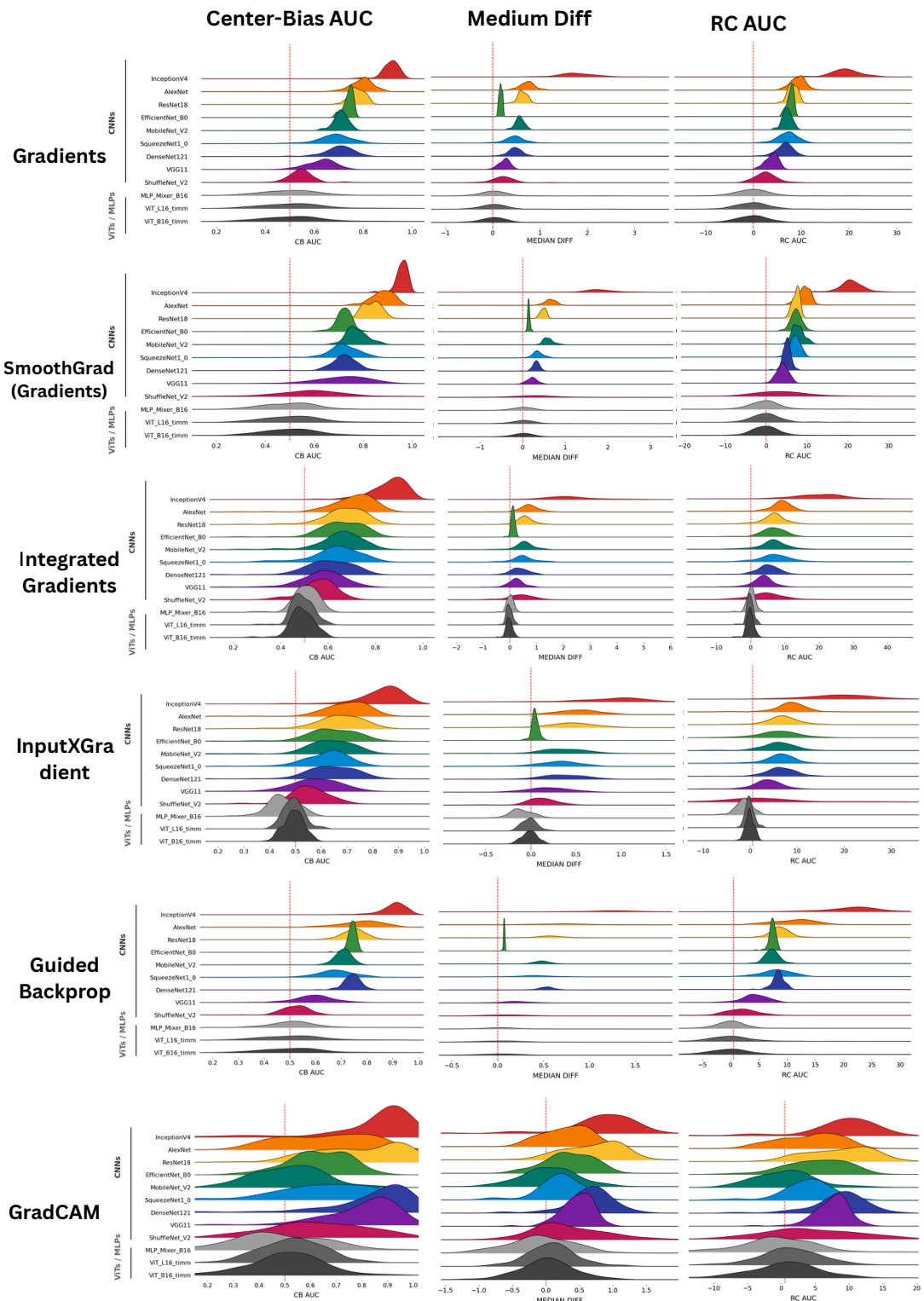

*Figure 12.* **Ridgeline distributions of center-bias metrics across architectures and saliency methods.** Rows correspond to six saliency methods: Gradients, SmoothGrad, Integrated Gradients, Input×Gradient, Guided Backprop, and GradCAM. Columns report three center-bias metrics: Center-Bias AUC (left), Median Difference (middle), and RC AUC (right). Dashed red lines indicate the corresponding no-bias baselines. Across methods, CNNs consistently exhibit stronger center bias than ViTs and MLP-based architectures.

Figure 12 extends the center-bias analysis to additional saliency methods. Across the five input-gradient-based methods (Gradients, SmoothGrad, Integrated Gradients, Input×Gradient, and Guided Backprop), we observe highly consistent architecture-level trends: CNN architectures exhibit strong positive center bias, whereas ViTs and MLP-based models remain near the no-bias baseline across metrics.

GradCAM exhibits broadly similar behavior, with most CNN architectures still showing positive center bias relative to ViTs and MLPs. However, the effect is generally weaker and more variable compared to input-gradient methods. This is expected because GradCAM operates on intermediate feature maps and depends on the choice of target convolutional layer, whereas our proposed mechanism specifically analyzes gradient propagation to the input. Despite these differences, the overall architectural trend remains consistent: convolutional architectures exhibit substantially stronger center-focused saliency structure than transformer- and MLP-based models.

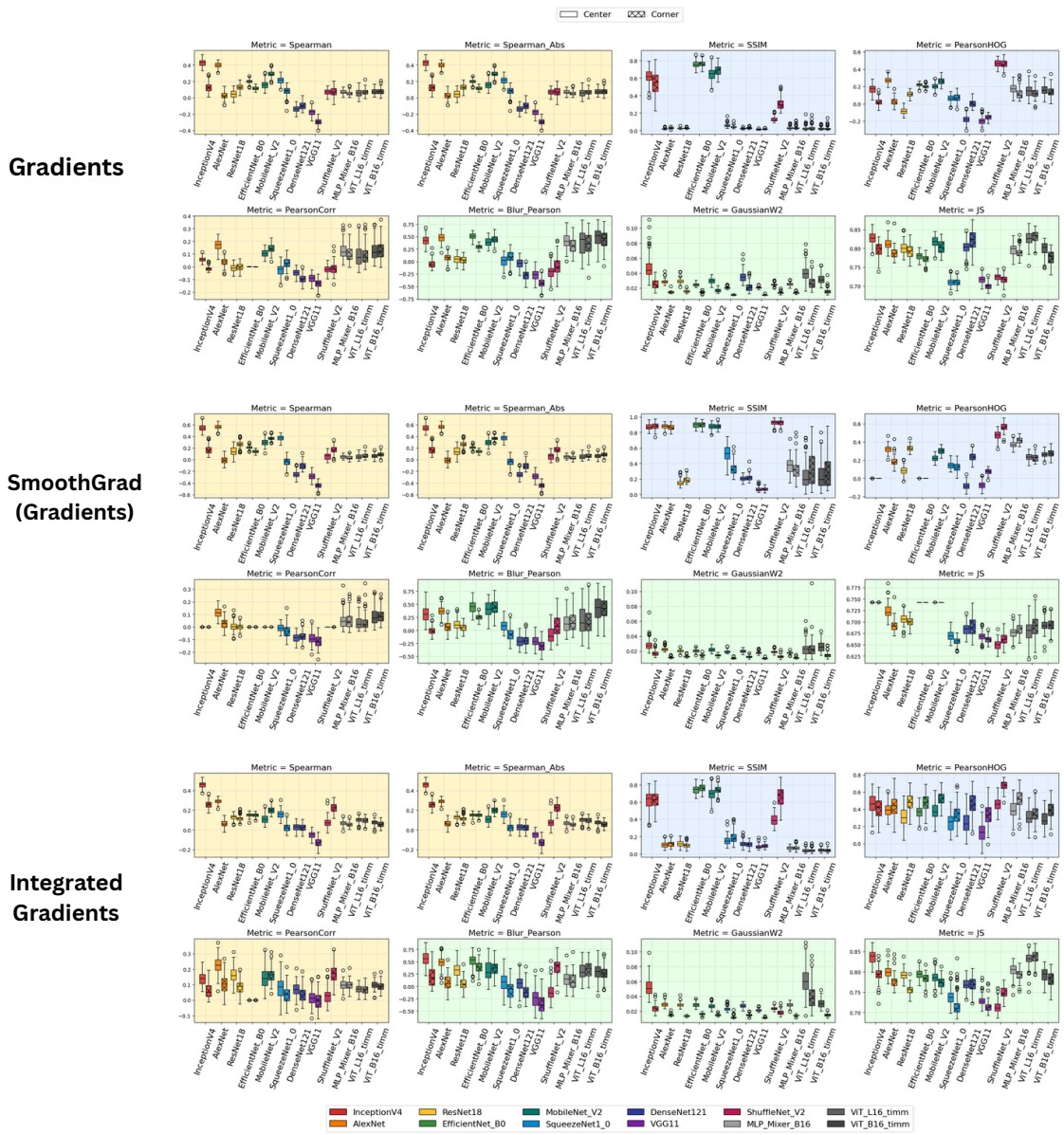

*Figure 13.* **Trained–random saliency similarity across attribution methods and metrics.** We report trained–random saliency similarity for three gradient-based attribution methods: vanilla gradients, SmoothGrad applied to gradients, and Integrated Gradients. Similarity is evaluated using eight metrics and aggregated across a population of images for multiple architectures included the main text, with center and corner object placements shown separately. While the main text focuses on vanilla gradients and a subset of representative metrics, these supplementary results show that the qualitative patterns—including elevated trained–random similarity under center-aligned inputs in two CNNs ()—are consistent across attribution methods and similarity measures.

## E. Architectural Center Bias Metrics

All metrics operate on the normalized saliency map $\tilde{A}(x) \in \mathbb{R}^{H \times W}$ defined in the main text, where $\sum_{i,j} \tilde{A}_{ij}(x) = 1$.

**Center-Bias AUC.** Let $\mathcal{C}, \mathcal{K} \subset \{1, \ldots, H\} \times \{1, \ldots, W\}$ denote the center and corner pixel index sets defined by fixed square masks. For a given image $x$, the Center-Bias AUC is

$$\text{CB-AUC}(x) = \Pr\left(\tilde{A}_{IJ}(x) > \tilde{A}_{UV}(x) \mid (I, J) \sim \text{Unif}(\mathcal{C}), \, (U, V) \sim \text{Unif}(\mathcal{K})\right).$$

In practice, this probability is estimated using the Mann–Whitney $U$ statistic applied to the samples $\{\tilde{A}_{ij}(x) : (i,j) \in \mathcal{C}\}$ and $\{\tilde{A}_{ij}(x) : (i,j) \in \mathcal{K}\}$.

**Median Difference.** For each image $x$, we define the median-based contrast

$$\Delta_{\mathrm{med}}(x) = \frac{\mathrm{median}\big(\tilde{A}_{\mathcal{C}}(x)\big) - \mathrm{median}\big(\tilde{A}_{\mathcal{K}}(x)\big)}{\mathrm{median}\big(\tilde{A}(x)\big) + \varepsilon},$$

where $\varepsilon > 0$ is a small constant for numerical stability. This statistic emphasizes typical saliency magnitudes within each region and is less sensitive to extreme values than rank-based measures.

**Radial Cumulative AUC (RC-AUC).** Let $c = ((H-1)/2, (W-1)/2)$ denote the image center and $d(i,j) = \|(i,j)-c\|_2$ the Euclidean distance of pixel $(i,j)$ from $c$. For image $x$, define the cumulative radial saliency mass

$$F_x(r) = \sum_{(i,j):\, d(i,j) \leq r} \tilde{A}_{ij}(x).$$

We compute RC-AUC as the signed area between $F_x(r)$ and the cumulative radial mass $F_{\mathrm{uni}}(r)$ of a spatially uniform distribution on the same grid:

$$\mathrm{RC\text{-}AUC}(x) = \int \big(F_x(r) - F_{\mathrm{uni}}(r)\big)\, dr.$$

Positive values indicate faster-than-uniform accumulation of saliency mass near the image center.

**Summary.** Across all three metrics (Fig. 12), convolutional networks exhibit a consistent and substantial center bias, even under sanity-check settings. In contrast, Vision Transformers remain tightly concentrated around the null baseline, indicating the absence of an architectural center preference. We exclude MLP-Mixer from this analysis because its gradient-based saliency maps are frequently degenerate under random initialization, making spatial statistics unreliable.

# F. Origin of Architectural Center Bias

This appendix provides two controlled ablation studies that isolate the architectural sources of center bias described in Section 3.2. All experiments here use randomly initialized networks unless stated otherwise, and therefore probe architectural effects independent of learned structures. The goal is to understand how spatial bias arises from convolutional design choices alone.

### F.1. Padding-Induced Variance Asymmetry

**Setup:** We study a shallow convolutional network with three convolutional-pooling blocks followed by global average pooling layer and a linear classifier. It is designed to isolate the effect of padding independent of depth and training. Each convolution uses a $3 \times 3$ kernel with stride 1 and padding of one pixel, followed by a ReLU nonlinearity. The three blocks have $3 \rightarrow 32 \rightarrow 64 \rightarrow 128$ channel dimensions, and $2 \times 2$ max pooling is applied after the first and second blocks. The feature map is reduced using adaptive global average pooling and mapped to a 10-way linear output layer. All models operate on $224 \times 224$ RGB inputs and are randomly initialized.

We vary only the padding choice used in all convolutional layers, considering four padding modes, zero, reflect, replicate, and circular padding, holding other design choices fixed.

For each padding mode, we initialize the model under multiple random seeds and compute input-gradient saliency maps for 100 ImageNet images. Saliency is defined as the $l_2$ norm of the gradient across the three color channels. We quantify spatial bias using the same Center-Bias AUC, median difference, and RC-AUC metrics used in the main body.

**Result:** Figure 14 shows that zero padding produces a consistent center bias under random initialization. Replacing zero padding with reflect, replicate, or circular padding reduces this effect across all metrics. The reduction is stable across seeds and images.

**Padding-Induced Architectural Bias Robustness**

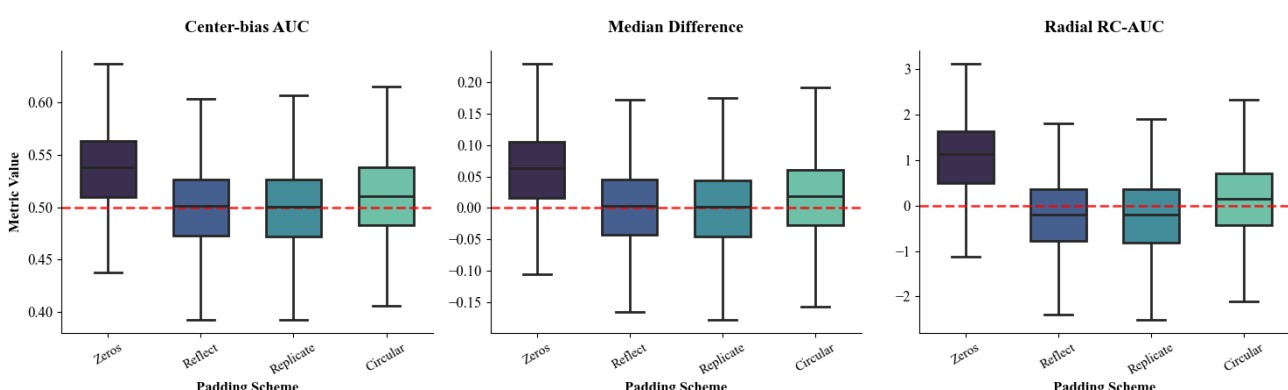

*Figure 14.* **Padding-induced architectural center bias across multiple metrics.** Comparison of randomly initialized CNNs using different convolutional padding schemes (zero, reflect, replicate, circular), evaluated with three complementary center-bias metrics: center-bias AUC (left), median center–corner difference (middle), and radial RC-AUC (right). Zero padding consistently induces stronger center bias across all metrics, while alternative padding schemes substantially reduce this effect. Red dashed lines indicate the null baseline (0.5 for AUC; 0 for difference-based metrics). Results are aggregated across multiple random seeds and input images.

**Interpretation:** This experiment directly supports the variance asymmetry argument in the main text. With zero padding, boundary locations systematically receive fewer nonzero inputs, leading to reduced activation and gradient variance near the image edges. Alternative padding schemes remove this boundary-induced signal loss, weakening the spatial prior. Note that no training is involved in this experiment: the effect arises entirely from how convolution interacts with padding choices.

This ablation shows that center bias is not an unavoidable property of convolution, but a consequence of a specific and widely used design choice.

### F.2. Depth, Receptive Field Growth, and Gradient Concentration

**Setup:** We next study how center bias evolves with depth. We construct a family of convolutional networks with increasing numbers of convolutional blocks, where each block of network consistsof a $3 \times 3$ convolution with stride 1 and zero padding, followed by a ReLU.The number of output channels grows with depth as $\min(32 \cdot 2^{d-1}, 256)$. A $2 \times 2$ max pooling operation is applied after each block as long as the spatial resolution remains greater than $1 \times 1$. Once the resolution collapses, the additional blocks consist only of convolution and ReLU. All models take 224X224 ImageNet inputs and use zero padding throughout. we compute the theoretical effective receptive field (RF) size by accounting for both convolutional kernel size and the pooling-induced stride growth. Specifically, the receptive field is updated recursively as $\mathrm{RF}_{\ell+1} = \mathrm{RF}_\ell + (k-1) \prod_{j \leq \ell} s_j$, where $k = 3$ is the convolutional kernel size and $s_j$ is the effective stride introduced by previous pooling layers. We stop increasing the stride once the spatial resolution reaches $1 \times 1$. Saliency maps and center-bias metrics are computed using the same procedure as in Appendix F.1.

**Results:** Center bias increases monotonically with depth as the effective receptive field expands. However, this increase saturates when the receptive field size exceeds the input dimensions. Beyond this point, additional depth produced little change in center bias scores (Figure 2b).

**Interpretation:** This behavior is consistent with the effective receptive field becoming more Gaussian-shaped as depth grows. Central pixels aggregate through more computational paths than peripheral pixels, leading to larger expected gradients. Once the RF covers the entire image, further increase in depth no longer adds to spatial preference, producing the observed plateau in Fig 2b.

This experiment explains why architectural center bias strengthens with depth but does not grow unlimitedly, and why very deep CNNs often show similarly center-focused saliency profiles despite different layer counts.

# G. Additional Experimental Details for the Corner-Shift Benchmark

## G.1. Training Setup and Data Generation

All models in Section 5 were trained on synthetic variants of the ImageNet training set, constructed using either a center-shrunk or corner-shrunk transformation. For both variants, the original image is resized and cropped to $224 \times 224$, after which the foreground object is uniformly downscaled by a factor of $0.5$ and embedded into a shuffled-background image (Figure 15).

**Center-shrunk images.** For the center-shrunk condition, the downscaled object is placed at the center of the image, with the remaining background filled by randomly permuted pixels from the original image to preserve the original color distribution.

**Corner-shrunk images.** For the corner-shrunk condition, the same downscaled object is placed in the top-left corner, with the remaining background similarly randomized. Both transformations preserve the original class label.

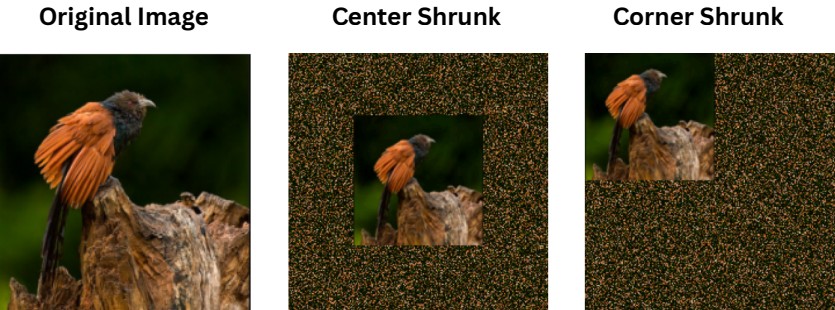

*Figure 15.* Illustration of the spatial transformations used to construct the corner-shift benchmark. **Left:** Original ImageNet image. **Middle:** Center-shrunk transform, where the object is uniformly downscaled and placed at the image center, with the remaining background filled by randomly permuted pixels. **Right:** Corner-shrunk transform, where the same downscaled object is placed in the top-left corner using an identical background randomization. Both transforms preserve the original class label and differ only in object location.

**Training protocol.** All models are initialized from ImageNet-pretrained weights and fine-tuned on the corresponding center-shrunk or corner-shrunk dataset for 40 epochs. Training is performed using stochastic gradient descent with a fixed batch size of 128. The final classification layer is replaced to match the 1000 ImageNet classes. Learning rates are selected per architecture based on preliminary tuning and are held fixed throughout training.

**Architectures.** We evaluate AlexNet, VGG11, ResNet18, DenseNet121, EfficientNet-B0, MobileNet-V2, ShuffleNet-V2, SqueezeNet1_0, and Inception-v4, as well as non-convolutional baselines including ViT-B/16, ViT-L/16, and MLP-Mixer-B/16. All models are initialized from ImageNet-pretrained weights and fine-tuned on the constructed center-shrunk and corner-shrunk datasets. Training hyperparameters follow standard ImageNet fine-tuning practices, with learning rates selected per architecture based on validation performance and held fixed across both spatial conditions.

## G.2. Classification Performance on the Corner-Shift Benchmark

To verify that models successfully learn the classification task under both spatial conditions and to ensure that incomplete saliency shifts are not attributable to training failure, we report Top-1 classification accuracy on the ImageNet validation set for all center-shrunk and corner-shrunk models in Table 1. Across architectures, all models achieve non-trivial validation accuracy, and performance is comparable between center-shrunk and corner-shrunk training. This confirms that CSI values below the idealized maximum reflect residual architectural bias rather than insufficient optimization or task failure.

*Table 1.* Top-1 classification accuracy (%) on the corner-shift benchmark. Models are fine-tuned from ImageNet-pretrained weights. Center-shrunk and corner-shrunk accuracies are comparable across architectures, indicating that observed saliency differences are not driven by training failure.

| Architecture | Center-shrunk | Corner-shrunk |
|---|---|---|
| AlexNet | 45.45 | 44.75 |
| VGG11 | 58.75 | 58.60 |
| ResNet18 | 50.89 | 50.86 |
| DenseNet121 | 62.73 | 62.04 |
| EfficientNet-B0 | 64.70 | 64.51 |
| MobileNet-V2 | 56.26 | 56.61 |
| ShuffleNet-V2 | 30.35 | 30.75 |
| SqueezeNet1_0 | 47.06 | 46.65 |
| Inception-v4 | 68.18 | 66.00 |
| MLP-Mixer-B/16 | 65.89 | 67.39 |
| ViT-B/16 | 74.59 | 75.53 |
| ViT-L/16 | 78.35 | 78.91 |

## H. Robustness of the Center Bias Metric and Corner Shift Benchmark

### H.1. Robustness to Crop Size

The Center-Bias AUC metric depends on the definition of the center and peripheral regions. In the main text, we use a center crop occupying one-half of the image width and height, together with four corner regions occupying one-quarter of the image width and height (Figure 3). To assess sensitivity to this design choice, we recomputed architecture-level center-bias rankings across a range of crop fractions.

Specifically, we varied the center-region fraction from 0.3 to 0.7 of the image width/height and recomputed mean Center-Bias AUC scores across architectures. We then compared the resulting architecture rankings against the default setting (0.5) using both Pearson and Spearman correlation.

Table 2 shows that the rankings are highly stable across crop sizes, with Spearman rank correlations ranging from 0.972 to 1.000. This indicates that the observed architectural trends are not sensitive to the precise crop definition.

*Table 2.* Robustness of architecture-level center-bias rankings across crop sizes. Correlations are computed relative to the default crop fraction of 0.5.

| Crop Fraction | Pearson Correlation | Spearman Correlation |
|---|---|---|
| 0.3 | 0.9952 | 0.993 |
| 0.4 | 0.9979 | 0.993 |
| 0.5 | 1.0000 | 1.000 |
| 0.6 | 0.9968 | 1.000 |
| 0.7 | 0.9818 | 0.972 |

### H.2. Robustness to Object Scale and Placement

The corner-shift benchmark is intended as a controlled spatial stress test that decouples architectural center priors from the spatial distribution of objects in the training data. To evaluate whether the observed effects depend on a specific benchmark construction, we performed additional robustness experiments varying both object scale and object placement.

Using ResNet18, we generated benchmark variants with object scales of 0.4, 0.5, and 0.6 relative to the original image size, and evaluated both top-left and bottom-right placements. For each configuration, we computed the Center-Shift Index (CSI).

Table 3 summarizes the results. Across all shifted-object configurations, training consistently produces large saliency displacements toward the object location, with CSI values remaining close to the expected geometric reference under perfect alignment. Centered-object controls remain near zero across scales.

*Table 3.* CSI metrics across different architecture configurations. Asterisks (*) denote the default settings used in the original paper.

| Placement | Scale | Mean CSI $\pm$ SD | Expected CSI | Normalized CSI |
|---|---|---|---|---|
| Center (*) | 0.5 | 0.064 $\pm$ 0.038 | 0.000 | – |
| Center | 0.4 | 0.050 $\pm$ 0.030 | 0.000 | – |
| Center | 0.6 | 0.084 $\pm$ 0.050 | 0.000 | – |
| Top-left (*) | 0.5 | 0.457 $\pm$ 0.047 | 0.502 | 0.910 |
| Bottom-right | 0.5 | 0.524 $\pm$ 0.052 | 0.502 | 1.044 |
| Top-left | 0.4 | 0.590 $\pm$ 0.039 | 0.605 | 0.974 |
| Top-left | 0.6 | 0.367 $\pm$ 0.068 | 0.404 | 0.909 |

These results indicate that the benchmark behavior is stable across different object scales and corner placements, and is not tied to a particular synthetic construction.

### H.3. Multiple Objects and Dataset-Level Object Distribution

Our analysis does not assume that each image contains a single centrally located object. Rather, the relevant quantity is the population-level spatial distribution of salient objects across the dataset.

To assess this distribution empirically, we analyzed the official ImageNet localization annotations using the same central benchmark region employed in our experiments (central $50\% \times 50\%$ crop). We find that approximately $76.8\%$ of annotated object centers lie within this region, and $92.5\%$ of images contain at least one centrally located object, despite many images containing multiple objects or complex scene structure. These statistics confirm a strong dataset-level tendency toward central object placement.

The corner-shift benchmark is designed to explicitly break this alignment between architectural center priors and dataset-level object statistics by controlling object placement while preserving object identity and background statistics. The resulting reduction in trained–random saliency similarity therefore reflects sensitivity to spatial alignment rather than an assumption of single-object images.

### H.4. Robustness Across Random Initializations

Figure 4 in the main text uses a single random initialization per architecture. To assess robustness to initialization, we repeated the random-initialization center-bias analysis using five random seeds for vanilla saliency. Table 4 shows that architecture-level Center-Bias AUC rankings remain highly stable across seeds, with seed-to-seed variation substantially smaller than the differences observed across architectures.

*Table 4.* Robustness across different random seeds. We report the Mean $\pm$ Std of CB-AUC for each architecture.

| **Architecture** | **CB-AUC (Mean $\pm$ Std)** |
|---|---|
| InceptionV4 | 0.920 $\pm$ 0.006 |
| AlexNet | 0.793 $\pm$ 0.001 |
| ResNet18 | 0.753 $\pm$ 0.010 |
| EfficientNet_B0 | 0.730 $\pm$ 0.040 |
| MobileNet_V2 | 0.715 $\pm$ 0.007 |
| SqueezeNet1_0 | 0.696 $\pm$ 0.010 |
| DenseNet121 | 0.671 $\pm$ 0.011 |
| VGG11 | 0.625 $\pm$ 0.005 |
| ShuffleNet_V2 | 0.580 $\pm$ 0.017 |
| MLP_Mixer_B16 | 0.520 $\pm$ 0.014 |
| ViT_L16_timm | 0.514 $\pm$ 0.002 |
| ViT_B16_timm | 0.513 $\pm$ 0.001 |

## H.5. Robust Training and Architectural Bias

To further examine the interaction between training dynamics and architectural center bias, we compared standard fine-tuning and FGSM-based adversarial fine-tuning on ResNet18 under the center-/corner-shift benchmark. Across Saliency, SmoothGrad, and Integrated Gradients, both standard and adversarial training substantially shifted saliency toward the object location under corner-shifted inputs, while centered inputs remained near CSI $\approx 0$.

Adversarial training generally moved CSI values closer to the ideal geometric reference under the corner-shift benchmark, indicating that training can further modulate the observed saliency distribution. At the same time, the qualitative architectural trends remained consistent with the main paper: training reshapes saliency structure, while architectural spatial priors continue to influence the resulting attribution patterns.

| Method | Training | CSI (Center) | CSI (Top-Left) |
|---|---|---|---|
| Saliency | Standard | -0.008 | 0.387 |
| Saliency | Adversarial | -0.017 | 0.483 |
| SmoothGrad | Standard | -0.017 | 0.464 |
| SmoothGrad | Adversarial | -0.022 | 0.477 |
| IG | Standard | -0.010 | 0.428 |
| IG | Adversarial | 0.005 | 0.496 |

*Table 5.* CSI under standard and adversarial fine-tuning on the corner-shift benchmark. Ideal top-left alignment corresponds to CSI $\approx 0.502$.

# I. Architectural Center Bias Metrics

Throughout this appendix, let $S \in \mathbb{R}^{H \times W}$ denote a normalized saliency map, i.e., $S_{ij} \geq 0$ and $\sum_{i,j} S_{ij} = 1$. We define a central region $\mathcal{C}$ and a peripheral region $\mathcal{K}$ consisting of four corner regions, as described in the main text.

**Center-Bias AUC.** The Center-Bias AUC measures the extent to which saliency values in the center tend to exceed those in the periphery. It is defined as the empirical proportion of center–corner pixel pairs for which the center saliency is larger:

$$\text{CB-AUC} = \frac{1}{|\mathcal{C}| \, |\mathcal{K}|} \sum_{(i,j) \in \mathcal{C}} \sum_{(u,v) \in \mathcal{K}} \mathbf{1}[S_{ij} > S_{uv}].$$

This score lies in $[0, 1]$, with $\text{CB-AUC} = 0.5$ indicating no systematic spatial preference. This statistic is equivalent to the Mann–Whitney $U$ statistic applied to the sets $\{S_{ij} : (i,j) \in \mathcal{C}\}$ and $\{S_{uv} : (u,v) \in \mathcal{K}\}$, but is presented here in this explicit form for interpretability and ease of comparison across region definitions.

**Radial Cumulative AUC (RC-AUC).** To capture global spatial concentration, we compute the cumulative radial mass

$$F(r) = \frac{\sum_{d(i) \leq r} S(i)}{\sum_i S(i)},$$

where $d(i)$ is the Euclidean distance from pixel $i$ to the image center. RC-AUC is defined as the signed area between the observed radial curve and a random baseline:

$$\text{RC-AUC} = \int \left( F_{\text{obs}}(r) - F_{\text{rand}}(r) \right) dr.$$

Positive values indicate excess saliency mass near the image center. RC-AUC is well suited to detecting smooth radial decay of saliency but is less directly aligned with discrete region-based stress tests, and its scale is not intrinsically bounded.

**Median Difference.** We additionally report the difference in median saliency between center and corner regions:

$$\Delta_{\text{med}} = \text{median}(S_{\mathcal{C}}) - \text{median}(S_{\mathcal{K}}),$$

optionally normalized by the global saliency scale. This metric provides a robust order-statistic-based summary but collapses each region to a single quantile and therefore does not capture differences in distributional shape or spatial concentration.

**Comparison of metrics.** Center-Bias AUC compares full saliency distributions across regions, is bounded between $0$ and $1$ with a clear no-bias reference at $0.5$, and is flexible with respect to region definitions, making it well matched to the region-based benchmarks used in the main text. RC-AUC and the median difference provide complementary views of center-focused behavior and are reported in the appendix as robustness checks and to provide a broader characterization of spatial saliency patterns.

## J. Definition of Sanity Check and Location-Sensitive Similarity Metrics

In this section we provide formal definitions of the similarity metrics used in our analysis. Following prior work on cascading randomization (Adebayo et al., 2018), we include several *structure-based* metrics (e.g. SSIM, HOG), but we show that these metrics are insufficient for detecting spatial saliency displacement. We therefore introduce a family of *location-sensitive* metrics designed to capture spatial correspondence in saliency maps.

Let $S, T \in \mathbb{R}^{H \times W}$ denote two normalized saliency maps (as defined in Section 3.1), assumed to be non-negative. For 3-channel saliency (e.g. guided backprop), we first convert the maps to grayscale by averaging across channels before computing spatial metrics.

### J.1. Pixelwise and Structure-Based Sanity Check Metrics

**Pixelwise Spearman Rank Correlation.** This metric is used in Adebayo et al. (2018) to measure the monotonic relationship between two saliency maps at the pixel level. Let $\text{vec}(\cdot)$ denote flattening into a vector. The Spearman correlation is

$$\rho_{\text{Spearman}}(S, T) = \text{corr}\big(\text{rank}(\text{vec}(S)), \text{rank}(\text{vec}(T))\big).$$

Because Spearman correlation compares ordinal structure at each pixel, it is insensitive to absolute magnitude but remains dominated by local pixel-level variability.

**Pearson Correlation.** Spatial correlation computes the Pearson correlation between flattened saliency maps:

$$\text{PearsonCorr}(S, T) = \text{corr}(\text{vec}(S), \text{vec}(T)).$$

Unlike rank-based Spearman correlation, this metric is sensitive to saliency magnitude, but it does not explicitly encode spatial neighborhood structure.

**SSIM (Structural Similarity Index).** SSIM is widely used in the sanity check literature to assess the similarity of local image patches. SSIM decomposes similarity into luminance, contrast and structure terms:

$$\text{SSIM}(S, T) = \frac{(2\mu_S \mu_T + C_1)(2\sigma_{ST} + C_2)}{(\mu_S^2 + \mu_T^2 + C_1)(\sigma_S^2 + \sigma_T^2 + C_2)}.$$

Although SSIM captures local texture consistency, it is *not* sensitive to global spatial displacement of saliency.

**HOG Feature Similarity.** Histogram of Oriented Gradients (HOG) descriptors capture local edge orientation patterns. Let $h(S)$ and $h(T)$ denote HOG feature vectors extracted from saliency maps $S$ and $T$. We define:

$$\text{HOGSim}(S, T) = \text{corr}(h(S), h(T)),$$

where $\text{corr}$ denotes Pearson correlation. HOG similarity emphasizes local edge-orientation statistics and is relatively insensitive to global spatial shifts in saliency mass, provided local structure is preserved.

### J.2. Location-Sensitive Similarity Metrics

The preceding metrics capture structural or textural similarity but fail to detect *where* saliency appears in the image. To quantify spatial alignment, we introduce four complementary metrics that explicitly measure positional correspondence.

**Gaussian-Blurred Pearson Correlation.** Blurring removes high-frequency texture while preserving coarse spatial mass. Let $\mathcal{G}_\sigma(\cdot)$ denote Gaussian smoothing with standard deviation $\sigma$. We define:

$$\mathrm{BlurPearson}(S, T) = \mathrm{corr}(\mathrm{vec}(\mathcal{G}_\sigma(S)), \mathrm{vec}(\mathcal{G}_\sigma(T))).$$

In practice, we implement $\mathcal{G}_\sigma$ using a Gaussian filter with $\sigma = 8$ pixels (fixed across all experiments). This choice smooths out fine edge-level details while retaining the overall spatial pattern of saliency, so the metric mainly shows whether salient regions appear in similar locations rather than whether they share the same exact local texture.

**Jensen–Shannon Spatial Divergence.** Define the Jensen–Shannon divergence between $P$ and $Q$:

$$\mathrm{JS}(P, Q) = \frac{1}{2} D_{\mathrm{KL}}(P \| M) + \frac{1}{2} D_{\mathrm{KL}}(Q \| M), \qquad M = \frac{1}{2}(P + Q).$$

We convert this into a similarity score using

$$\mathrm{JSSim}(S, T) = \frac{1}{1 + \mathrm{JS}(P, Q)}.$$

JS divergence penalizes differences in saliency mass allocation across the image.

**Gaussian Wasserstein-2 Similarity.** To capture global spatial differences in saliency while accounting for both location and spread, we approximate each normalized saliency map by a two-dimensional Gaussian distribution with matched first and second moments. Specifically, for a normalized saliency map $S$, we compute its spatial mean $\mu_S$ and covariance $\Sigma_S$, and define the squared Wasserstein-2 distance between two maps $S$ and $T$ as

$$W_2^2(S, T) = \|\mu_S - \mu_T\|_2^2 + \mathrm{Tr}\left(\Sigma_S + \Sigma_T - 2\left(\Sigma_T^{1/2} \Sigma_S \Sigma_T^{1/2}\right)^{1/2}\right).$$

We convert this distance into a similarity score via $\mathrm{GaussianW2Sim}(S, T) = 1/(1 + W_2(S, T))$. Unlike pixelwise or structure-based metrics, Gaussian W2 is sensitive to global saliency displacement and concentration, making it well suited for detecting center-focused saliency patterns induced by architectural priors.

**Interpretation of Distributional Metrics.** Distribution-based similarity measures such as Jensen–Shannon (JS) divergence and Gaussian Wasserstein distance compare global saliency mass allocation rather than precise spatial alignment. As a result, JS similarity is often high across models when saliency distributions share similar overall spread, even if their mass is concentrated in different locations. Gaussian Wasserstein similarity, which incorporates both mean location and spatial dispersion via second-order moments, is more sensitive to center-focused saliency and therefore tends to highlight architectural center bias more strongly. These behaviors are consistent with the trends observed in Figure 11.

## K. Qualitative Examples

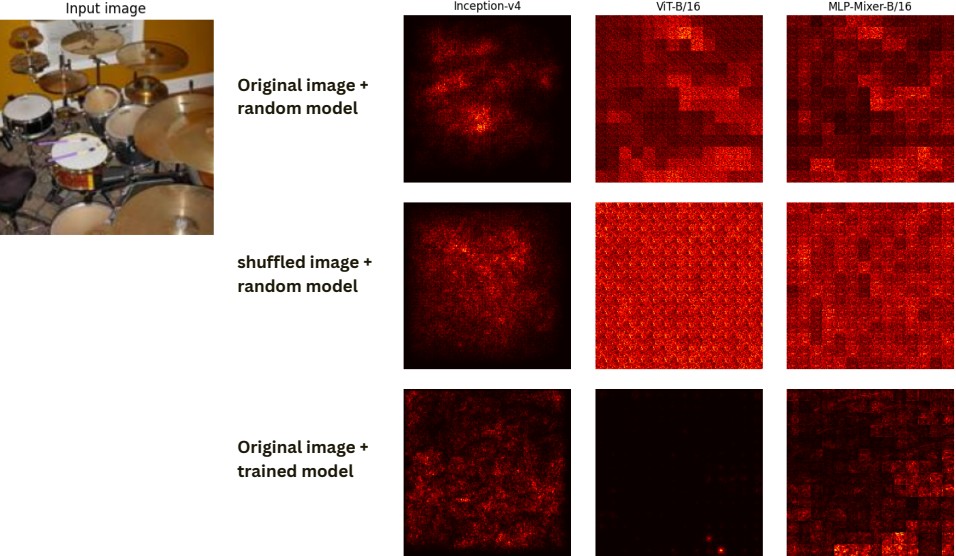

*Figure 16.* **Architecture-dependent saliency patterns.** Qualitative saliency maps for a CNN (Inception-v4), a vision transformer (ViT-B/16), and an MLP-based model (MLP-Mixer-B/16) under three conditions: original image with a random model, pixel-shuffled image with a random model, and original image with a trained model (rows, top to bottom). The input image is shown on the left for reference.

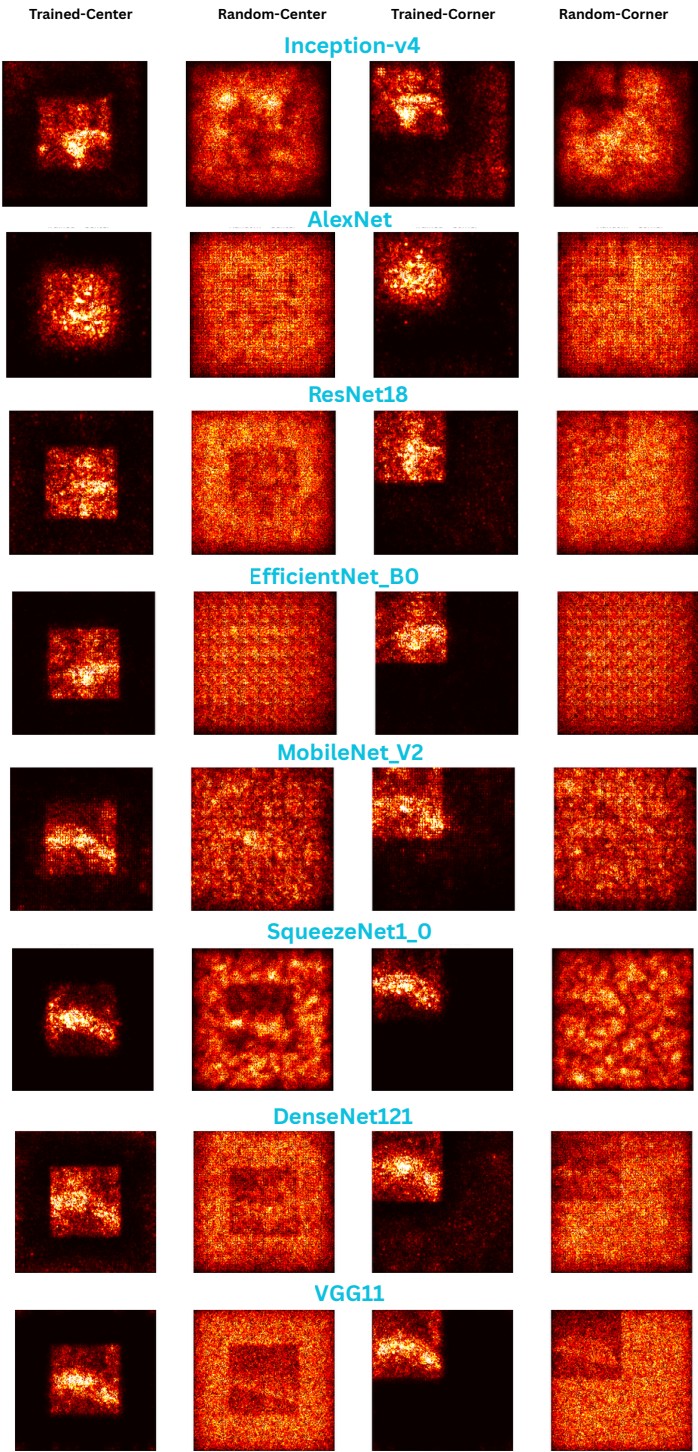

*Figure 17.* **Qualitative saliency examples under center- and corner-shifted inputs (CNNs).** Saliency maps for representative convolutional architectures under four evaluation conditions: trained model with centered inputs, random model with centered inputs, trained model with corner-shifted inputs, and random model with corner-shifted inputs (columns, left to right). Each row corresponds to a different CNN architecture.

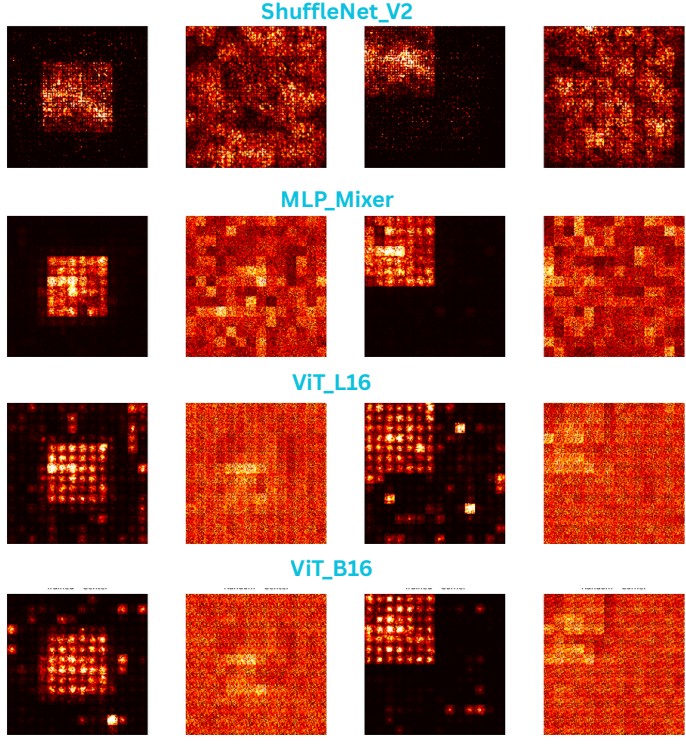

*Figure 18.* **Qualitative saliency examples under center- and corner-shifted inputs (ShuffleNet, ViTs and MLPs).** Same layout as the previous page, showing transformer-based and MLP-based architectures. Columns correspond to trained versus random models under centered and corner-shifted inputs.

