# OpenReview forum: "When Random Saliency Looks Trained: Architectural Center Bias in CNN Interpretability"
_ICML.cc/2026/Conference — ICML 2026 regular_

### Official Review · Reviewer_CZN4 · 2026-03-10

**Soundness:** 3
**Presentation:** 3
**Significance:** 2
**Originality:** 2
**Overall Recommendation:** 3
**Confidence:** 4

**Summary:**

The paper discusses how  saliency , a common  method  used for explainability of neural networks, may show enhanced results in CNNs due to an architectural bias, namely center bias, that causes that CNNs trained under center bias dataset show higher activation in objects that are centered, not necessarily because of training but because of an artifact introduced by zero padding causing that pixels in the borders have less variance than those pixels from the center. The paper also introduces a benchmark that shift the objects to corners disentangling the possible confound.

**Compliance With Llm Reviewing Policy:**

Affirmed.

**Key Questions For Authors:**

Included in the weakness section

**Limitations:**

yes

**Strengths And Weaknesses:**

Strenghts
The paper is introduced clearly and cleanly.  There is an interesting  empirical  and theoretical build up to justify the results and attempt  provide a mechanistic justification of the phenomenon.

Weakness

The main weakness of the paper may rely on the impact. Saliency has been long known to have several weakness, one of them that networks initialized with random weights  seem to generate similar saliency maps to those networks trained to perform certain tasks. Now, the paper goes further to clarify why that may be, but I am unsure on how to apply the result? Does CNNs trained on centered images, failed on those objects presented on the corner?  Or should we apply this result as a way to evaluate saliency as a method for explaining CNNs? In that case, comparisons with other explainability methods that were not included should be also performed , would occlusion based methods fail for the same architectural bias?
How this method can explain or guide into possible failure modes for networks?


Would this center bias be addressed by other training methods? For example robust trained networks?

Figure 3 : Axis are all over the place, which may obscure a bit the result.
Figure 6: Some images do not have y axis ticks, is this intentional?

---

> ### Author Rebuttal · Authors · 2026-03-31
>
> We appreciate the reviewer’s thoughtful feedback and helpful questions on impact and scope. We clarify the practical implications and strengthen empirical support as follows:
>
> - **Practical impact. how to use the result.** Our main implication is for evaluating and interpreting saliency methods. We show that commonly used similarity-based sanity checks can be systematically inflated by architectural priors (e.g., center bias), even when saliency reflects learned structure. This suggests that such evaluations for explanation reliability can be confounded by such priors.
>   Concretely, our results motivate simple diagnostic intervention, such as spatial stress tests (e.g., object relocation), to disentangle learned behavior from architectural bias in practice. This provides a practical tool for more reliable evaluation of interpretability methods.
> - **Additional explainability methods.** Our mechanism is derived for gradient-based saliency methods, where gradient propagation and activation variance are central. To strengthen this point, we extended experiments to additional gradient-based methods (Guided Backprop and Input×Gradient), and observe strong agreement (Spearman ρ ≈ 0.91) with consistent architectural trends (CNNs center-biased, ViTs/MLPs near neutral).
>    In addition, benchmark results are consistent across methods: under corner-shift training, CSI shifts toward the scale-specific ideal (≈ 0.5) but not fully, and trained–random similarity patterns across architectures remain stable.
>   This indicates that the observed effect persists across the gradient-based family and is not specific to a single explainer.
>    Methods such as occlusion or CAM-based approaches follow different principles and are outside the scope of the current mechanism; we consider systematic study of these families as future work.
> - **Mitigation and training considerations.** Our results suggest that center bias arises primarily from architectural design (e.g., padding and receptive field growth), and may therefore persist across training regimes. However, our ablations show that alternative padding schemes (e.g., circular padding) can reduce this bias, suggesting a concrete architectural direction for mitigation. Studying alternative training strategies (e.g., robust training) is an interesting future direction.
> - **Figures.** Thank you for pointing this out.
>     For Figure 3, the differing axis ranges are intentional: different architectures produce saliency maps with different overall scales, and we focus on comparing distributional overlap rather than absolute values.
>     For Figure 6, some panels share axes for visual alignment across conditions, which results in missing ticks in certain rows. We will clarify this in the caption to improve readability.

---

> > ### Author Rebuttal · Reviewer_CZN4 · 2026-03-31
> >
> > I thank the authors for the response. I am not sure I agree with the CAM based approach argument, at least  for GradCAM. This method   is explicitly gradient-based: It computes gradients of the class score with respect to feature map activations and then weights those maps accordingly. I think  this one should be included. Even if it does not exhibit center-bias it would be an interesting finding that warrant explanation.
> >
> > I am not so sure also to quickly dismiss robust training, I think it goes to heir central claim. If center bias arises purely from architectural properties like zero padding and receptive field growth, it should persist in robustly trained models regardless of the training objective. If it does not, and there is reason to suspect it might not, given that robust training is known to produce structurally different gradient landscapes,  then the paper's framing of center bias as an architectural prior independent of training would need to be properly discussed. I would not say this is a peripheral extension but a direct empirical test of the paper's main thesis, which would greatly strength the paper.

---

> > > ### Author Response · Authors · 2026-04-04
> > >
> > > We thank the reviewer for the helpful follow-up questions and suggestions.
> > > - **Grad-CAM.** We agree that Grad-CAM should be included. We therefore added Grad-CAM analysis using a consistent late-convolution layer across CNN architectures. Under this setup, most CNNs exhibit positive center bias (mean CB-AUC above 0.5), broadly consistent with our main finding, although the effect is generally weaker compared to input-gradient methods (vanilla, Integrated Gradients, SmoothGrad, Guided Backprop, and Input×Gradient). We also observe larger seed sensitivity for Grad-CAM (standard deviation up to approximately 0.17 for some models) and dependence on target layer choice. We will include these results and clarify this point in the revision.
> > > | Architecture    | Mean Center Bias | Std (across 5 seeds) |
> > > |-----------------|------------------|--------------------|
> > > | ResNet18        | 0.81             | 0.17               |
> > > | VGG11           | 0.78             | 0.01               |
> > > | DenseNet121     | 0.77             | 0.17               |
> > > | AlexNet         | 0.72             | 0.11               |
> > > | MobileNet_V2    | 0.65             | 0.03               |
> > > | SqueezeNet1_0   | 0.64             | 0.06               |
> > > | EfficientNet_B0 | 0.64             | 0.04               |
> > > | ShuffleNet_V2   | 0.58             | 0.08               |
> > > | InceptionV4     | 0.50             | 0.12               |
> > >
> > >
> > > - **Robust training / training effects.** Thank you for raising this point regarding robust training. We agree this is relevant to our main claim. We would like to clarify that we did not intend to claim that center bias remains regardless of training. We acknowledge that our previous wording (“exists independent of training”) may have caused confusion, and we will revise the paper to clarify this point.
> > >
> > >   Our intent was not to suggest that center bias remains unchanged under training. Rather, our results support that the **bias is present at initialization and arises from architectural design, while its observed strength is modulated by training**.
> > >
> > >   This is reflected in our corner-shift experiments, where training substantially shifts saliency toward the object location, demonstrating that training can modulate center bias. At the same time, this also illustrates that **without a stress test such as our corner-shift setting**, the effect of training cannot be clearly assessed, since object location and architectural center bias are otherwise confounded. Our experiments further indicate that residual spatial bias can remain, consistent with our interpretation that training reduces but does not fully eliminate the architectural effect.
> > >
> > >   Following the reviewer’s suggestion, we added an analysis of robust training: we compared standard and FGSM-based adversarial fine-tuning on ResNet18 under the same center-/corner-shift benchmark. In all three methods already used in the paper (Saliency, SmoothGrad, IG), the center condition remains near CSI = 0, while the top-left condition moves substantially toward the ideal CSI = 0.502. Adversarial training shifts the corner-trained model even closer to this ideal (Saliency: 0.387 → 0.483, SmoothGrad: 0.464 → 0.477, IG: 0.428 → 0.496), showing that robust training, like standard training, also modulates the observed saliency pattern in this setting.
> > >
> > >   | Method       | Training     | CSI (Center) | CSI (Top-Left) | Ideal (Top-Left) |
> > >   |--------------|-------------|--------------|----------------|------------------|
> > >   | Saliency     | Standard    | -0.008       | 0.387          | 0.502            |
> > >   | Saliency     | Adversarial | -0.017       | 0.483          | 0.502            |
> > >   | SmoothGrad   | Standard    | -0.017       | 0.464          | 0.502            |
> > >   | SmoothGrad   | Adversarial | -0.022       | 0.477          | 0.502            |
> > >   | IG           | Standard    | -0.010       | 0.428          | 0.502            |
> > >   | IG           | Adversarial | 0.005        | 0.496          | 0.502            |
> > >
> > >   We thank the reviewer for raising this point; we will include additional discussion of how different training schemes interact with architectural bias in the appendix.
> > >
> > > Overall, we will revise the paper to (i) include Grad-CAM under a consistent setup, and (ii) clarify that the paper’s claim is architectural origin with training interaction, and include discussion of additional training schemes in the appendix.

---

### Official Review · Reviewer_vRuw · 2026-03-12

**Soundness:** 3
**Presentation:** 3
**Significance:** 2
**Originality:** 3
**Overall Recommendation:** 3
**Confidence:** 4

**Summary:**

This paper studies a classical phenomenon in saliency research: why randomly initialized CNNs can produce saliency maps that look similar to those of trained models. The main claim is that this similarity should not be attributed entirely to the saliency method itself, but is partly caused by an architectural center bias in CNNs. Through center-bias measurements, controlled analysis, cross-architecture comparisons, and the proposed corner-shift benchmark with Center-Shift Index (CSI), the paper argues that training does move saliency toward the object location, but standard centered settings can also inflate trained–random saliency similarity because of architectural priors. Overall, the paper’s main message is that architectural priors can confound standard saliency evaluation.

**Compliance With Llm Reviewing Policy:**

Affirmed.

**Final Justification:**

I appreciate the authors’ detailed rebuttal, which has addressed part of my concerns. However, I am not sufficiently convinced to change my initial weak reject rating.

**Key Questions For Authors:**

1. Does the paper establish that center bias is an important factor, or that it is the dominant explanation for trained–random similarity?
2. How robust is the corner-shift benchmark under different corner placements, object scales, or background constructions?
3. Beyond CSI, could the paper include a more direct localization-quality metric?
4. To what extent does the proposed mechanism remain valid in more complex modern CNN designs?

**Limitations:**

yes

**Strengths And Weaknesses:**

Strengths
1. The problem is well motivated.
Rather than repeating the known claim that saliency methods may be unreliable, the paper asks where trained–random similarity actually comes from. This is a meaningful and well-scoped question.
2. The experimental story is coherent.
The paper moves from qualitative center-bias observations, to mechanism analysis, to cross-architecture comparison, and finally to the corner-shift benchmark. The overall logic is easy to follow.
3. The corner-shift benchmark is a useful idea.
It breaks the alignment between CNN center bias and object-centered data distributions, making the training effect easier to isolate.

Weaknesses
1. The paper shows that center bias affects trained–random similarity, but does not fully establish that it is the main source of the phenomenon. The current evidence more strongly supports the claim that center bias is an important confounder.
2. The corner-shift benchmark is better viewed as a stress test than as direct evidence from natural settings. Since it relies on relatively strong data construction, the extent to which the conclusions transfer to standard evaluation settings remains somewhat unclear.
3. CSI is useful for measuring whether saliency moves away from the center, but it is not by itself a complete measure of accurate target localization.
4. The mechanistic explanation is plausible and empirically supported, but it is not yet fully definitive; the paper does not completely rule out other architectural factors or quantify their relative contributions.

---

> ### Author Rebuttal · Authors · 2026-03-31
>
> We appreciate the review’s thoughtful and constructive feedback. We clarify the scope of our claims and strengthen empirical support as follows:
>
> - **Center bias as confounder (not sole explanation).** We agree that our results most directly establish architectural center bias as an important confounder rather than the dominant explanation. Our contribution is to identify and isolate this factor, and show that it can independently induce structured saliency patterns and inflate trained–random similarity, even in the absence of semantic structure (e.g., pixel-shuffled inputs, Fig. 1). This complements prior method-focused explanations (e.g., edge effects) by introducing an architectural perspective. We will clarify this positioning.
> - **Corner-shift benchmark as stress test + robustness.** We agree the benchmark is best viewed as a controlled stress test designed to decouple architectural priors from dataset biases. Importantly, center bias already inflates trained–random similarity under standard evaluation (Fig. 6), while corner-shift reveals this effect more clearly by breaking alignment.We further evaluated robustness across object scale and placement (ResNet18), and observe consistent behavior:
> | Position      | Scale | CSI (mean ± sd) | Ideal CSI | Normalized CSI |
> |---------------|-------|------------------|-----------|----------------|
> | Center (*)    | 0.5   | 0.064 ± 0.038    | 0.000     | –              |
> | Center        | 0.4   | 0.050 ± 0.030    | 0.000     | –              |
> | Center        | 0.6   | 0.084 ± 0.050    | 0.000     | –              |
> | Top-left (*)  | 0.5   | 0.457 ± 0.047    | 0.502     | 0.910          |
> | Bottom-right  | 0.5   | 0.524 ± 0.052    | 0.502     | 1.044          |
> | Top-left      | 0.4   | 0.590 ± 0.039    | 0.605     | 0.974          |
> | Top-left      | 0.6   | 0.367 ± 0.068    | 0.404     | 0.909          |
> (* = original paper setting)
>
>   Center controls remain near zero, while shifted-object training consistently produces large CSI close to scale-specific ideals. This shows the effect is stable across configurations and not tied to a specific construction.
>
>   We also note that architecture-level center-bias rankings are stable across crop sizes and settings (see Reviewer 1cJf response (crop size and robustness)), further supporting the robustness of this effect.
>
> - **CSI vs localization.** We agree CSI does not directly measure localization accuracy. Our goal is to quantify relative spatial shifts under controlled interventions.
> We explored localization metrics based on bounding boxes (e.g., saliency mass within object regions), but found them difficult to interpret in this setting: the scores depend strongly on object size and position, and higher values can arise either because saliency shifts toward the displaced object (in the corner condition) or simply because the object overlaps with the center bias (in the center condition). As a result, these metrics mix multiple effects and do not provide a clear or consistent signal in our benchmark.
>                       Instead, we use CSI as a targeted and interpretable measure of relative spatial displacement with a clear reference value.
>
> - **Mechanism validity in modern CNNs.** We agree the mechanism is not intended as a complete explanation. To assess generality, we conducted additional analyses on ResNet18 (standard) and **ConvNeXt-Tiny (modern CNN)** (will be included in appendix). Across both, we observe consistent **signatures predicted by our mechanism**: (1) higher variance near the center than at the boundaries, increasingly pronounced with depth, (2) an increasing center–periphery gap with depth, and (3) smooth radial decay rather than noise-like structure (see variance maps https://github.com/anonymousresearcher672/anonymous-project-2026/blob/main/variance_ac_depth.png and radial profiles https://github.com/anonymousresearcher672/anonymous-project-2026/blob/main/radial_profile.png ). These results show the effect persists in modern architectures, even if other factors may also contribute.

---

> > ### Author Rebuttal · Reviewer_vRuw · 2026-04-01
> >
> > The rebuttal clarifies the scope of the claims and provides additional empirical evidence, particularly regarding robustness across settings. However, these clarifications primarily improve the presentation and positioning of the work rather than fundamentally strengthening its core contribution.
> > In particular, the paper now frames center bias as an important confounder rather than a dominant explanation, which I agree is more appropriate. However, this also limits the overall significance of the contribution, as the work does not fully establish the relative importance of this factor compared to other explanations.
> > Furthermore, key concerns remain only partially addressed. The mechanistic explanation is still not fully definitive, and the extent to which the findings generalize beyond the controlled corner-shift setting remains unclear. While the additional analyses are helpful, they do not substantially change my assessment of the paper’s limitations.

---

> > > ### Author Response · Authors · 2026-04-04
> > >
> > > We thank the reviewer for the additional clarification and agree that the rebuttal helped sharpen the scope of our claims. We would like to take this opportunity to restate the main contribution and its practical impact more clearly.
> > >
> > > Our contribution is not to establish architectural center bias as the dominant explanation for trained–random similarity, but to show that it is a **systematic and previously unaccounted-for confounder** in widely used evaluation protocols. In this setting, even a single uncontrolled factor is sufficient to distort conclusions about explanation reliability, regardless of its relative contribution compared to other effects.
> > >
> > > We also clarify that the corner-shift setting is not intended to serve as a naturalistic benchmark requiring broad generalization, but as a **controlled intervention** designed to break the alignment between architectural priors and dataset biases. Its purpose is to study how training interacts with this architectural prior on saliency behavior, rather than to model real-world data distributions.
> > >
> > > Our work revisits a widely studied phenomenon in interpretability evaluations—trained–random saliency similarity, as highlighted in prior “sanity check” studies and subsequent work that continues to rely on similarity-based evaluation protocols. We show that this effect is not solely attributable to properties of the saliency method, but can arise from **architecture-induced spatial priors**.
> > >
> > > In particular, we demonstrate that randomly initialized CNNs exhibit structured center-focused saliency even under pixel-shuffled inputs, indicating that explanations based purely on edge or object structure are insufficient. We further show that this behavior is **architecture-specific** (strong in CNNs, weak in ViTs/MLPs), and provide **empirical and mechanistic evidence** linking it to standard convolutional design choices.
> > >
> > > Importantly, this has direct practical implications. We show that commonly used similarity-based sanity checks can be systematically confounded by architectural priors, which can inflate trained–random similarity even when models have learned meaningful structure (Fig. 6). This affects how saliency methods are evaluated and interpreted in practice.
> > >
> > > To address this, we propose simple **spatial stress tests** (e.g., object relocation) as a diagnostic tool to disentangle learned behavior from architectural bias and to reveal how similarity-based evaluations can be confounded. These interventions are easy to implement and provide a useful diagnostic signal for understanding how architectural bias influences similarity-based evaluation. In particular, CSI and similarity-based metrics capture complementary aspects: CSI quantifies how training redistributes saliency spatially, while similarity metrics reveal how architectural bias can confound evaluation.
> > >
> > > Overall, our goal is to complement prior method-centered explanations with an architectural perspective, and to provide both **evidence and practical tools** for more reliable evaluation of interpretability methods.

---

### Official Review · Reviewer_9TdE · 2026-03-13

**Soundness:** 3
**Presentation:** 3
**Significance:** 3
**Originality:** 3
**Overall Recommendation:** 4
**Confidence:** 4

**Summary:**

This paper presents a complementary perspective to the sanity checks for saliency maps introduced by Adebayo et al. (2018). The central claim is that architectural biases in CNNs, particularly network depth and the use of zero padding, induce a center bias in saliency maps.

Building on this observation, the authors argue that the commonly observed similarity between saliency maps from trained and randomly initialized networks does not necessarily imply that the explanation method is insensitive to model parameters. Instead, they suggest that this similarity can arise from the interaction between an architecture-induced center bias and the effect of training, which shifts spatial attributions toward object locations that are frequently centered in datasets such as ImageNet.

Empirically, the paper presents an extensive experimental study across 12 models, including CNNs, MLPs, and Vision Transformers, and performs several ablation experiments to evaluate the proposed sources of center bias.

**Compliance With Llm Reviewing Policy:**

Affirmed.

**Final Justification:**

This paper makes a useful contribution by revisiting the saliency-map sanity-check literature and arguing that CNN-induced center bias can be an important confounder in explaining why saliency maps may remain similar under trained and randomized models. I found this perspective interesting, and the empirical study is reasonably broad, covering multiple architectures and metrics.

The rebuttal addressed my main concerns to a meaningful extent, particularly by providing additional empirical evidence that strengthens the paper. This improved my confidence in the work, and I therefore lean to a weak accept.

That said, my enthusiasm remains limited after considering the concerns raised by other reviewers, particularly reviewer vRuw: the overall **significance** is somewhat modest as the paper offers an important refinement to the interpretation of prior sanity-check results, but it does not fully establish the relative importance of center bias compared with other possible explanations.

Overall, I view the paper as a **weak accept**. The core idea is interesting, the empirical analysis is solid enough to support publication, and the rebuttal addressed the main issues sufficiently, even though some limitations remain.

**Key Questions For Authors:**

- In Figure 5, why does the CSI take negative values for *Center Shrunk*? Based on its definition, it seems that this quantity should be nonnegative.
- For Figure 5, it would be useful to include a comparison with a model trained on standard ImageNet, without *Center Shrunk* or *Corner Shrunk* fine-tuning, and then evaluate its CSI on both *Center Shrunk* and *Corner Shrunk* inputs. This would help quantify more clearly how much the fine-tuning changes the behavior.
- Why is the CB-AUC metric defined using only the corner regions, rather than the full band surrounding the central block?
- Is Figure 4 based on a single random initialization for each model? If so, it would be helpful to report the same analysis over several random initializations in order to assess the stability of the effect.
- Do you think it would be more appropriate to do the corner-shift training from scratch, rather than starting from ImageNet-pretrained weights? Since pretraining on ImageNet may already induce a center bias due to object centering, fine-tuning alone may not be sufficient to fully remove this bias.
- In the experiments with randomly initialized models, are the weights sampled using He initialization, or from a standard Gaussian distribution such as $\mathcal{N}(0,1)$ (i.e., no fan scaling)?
- In Section 3.2, paragraph *From activation variance to gradient magnitude*, do you have experimental plots supporting this claim? Such evidence would considerably strengthen this part of the argument.
- Do you provide either references or empirical evidence supporting the claim that objects in ImageNet are typically centered in the image?

**Limitations:**

Yes but not clearly. It could be done in a section.

**Strengths And Weaknesses:**

**Strengths:**

- This paper provides a valuable extension of [1], offering a different perspective on the observation that saliency maps can remain similar for trained and randomized models. Rather than attributing this phenomenon solely to a failure of the explanation method, the paper argues that CNN-induced center bias can substantially contribute to the observed similarity.
- The experimental section is extensive and covers a broad range of architectures, including both CNNs and non-CNN baselines such as Vision Transformers and MLPs. The use of multiple evaluation metrics across experiments also strengthens the empirical analysis.

**Weaknesses:**

- **The related work discussion is limited and omits several relevant papers:** Work [2] identifies a related padding-induced bias in CNNs. Although that paper is not framed as an interpretability study, it still appears closely related and should be discussed, both for similarity and to better position the present contribution relative to prior findings. In addition, the paper does not discuss other works that, like this one, provide a complementary or additive perspective on the conclusions of [1], such as [4,5].
- The empirical study could be broadened to include more saliency methods. At present, the paper only evaluates three gradient-based explainers, while other relevant approaches exist, including [3] and at least the methods originally considered in [1]. It would also be useful to include saliency methods from other families, such as CAM-based approaches, to assess whether the proposed findings extend beyond fully-gradient-based explanations.
- No code appears to be provided, which limits the reproducibility of the work, especially given the scope of the empirical claims.
- More broadly, I have reservations about several aspects of the paper, which I detail in the questions below.

**Additional comments:**

- Typo in line 94: “emaphsis”.
- In Figure 6(d), the histogram labels are extremely small and difficult to read.
- Why does the following entry appear in the references: *Langley, P. Crafting papers on machine learning. In Langley, P. (ed.), Proceedings of the 17th International Conference on Machine Learning (ICML 2000), pp. 1207–1216, Stanford, CA, 2000. Morgan Kaufmann.*?

**References**:

[1] *Sanity Checks for Saliency Maps*, Julius Adebayo, Justin Gilmer, Michael Muelly, Ian Goodfellow, Moritz Hardt, Been Kim, Neurips 2018.

[2] *Mind the Pad -- CNNs can Develop Blind Spots*, Bilal Alsallakh, Narine Kokhlikyan, Vivek Miglani, Jun Yuan, Orion Reblitz-Richardson, ICLR 2021.

[3] *Full-Gradient Representation for Neural Network Visualization*, Suraj Srinivas, Francois Fleuret, Neurips 2019.

[4] *Shortcomings of Top-Down Randomization-Based Sanity Checks for Evaluations of Deep Neural Network Explanations*, Alexander Binder, Leander Weber, Sebastian Lapuschkin, Grégoire Montavon, Klaus-Robert Müller, Wojciech Samek, CVPR 2023

[5] *Revisiting Sanity Checks for Saliency Maps*, Gal Yona, Daniel Greenfeld, XAI4Debugging@NeurIPS2021.

---

> ### Author Rebuttal · Authors · 2026-03-31
>
> We thank the reviewer for their detailed and constructive feedback.
> - **Related work and scope of methods.** We thank the reviewer and will expand the related work. Prior work has primarily explained trained–random similarity via saliency methods: [1] shows many methods are insensitive to model parameters under randomization, suggesting similarity may arise from the explainer, and follow-ups [4,5] show such sanity checks can be misleading or inconclusive.
> Separately, [2] identifies padding-induced spatial bias in CNNs (reduced activation near boundaries), though not in the context of interpretability.
>   Our contribution complements these lines of work: rather than attributing similarity solely to saliency methods, we show that architecture-induced spatial bias can independently produce structured saliency patterns and inflate trained–random similarity, with a mechanism related to the padding effects in [2].
> - **Additional explainability methods.** We focus on gradient-based saliency, where the mechanism relates to gradient propagation and activation variance. CAM or perturbation methods rely on different mechanisms and are outside our scope. However, we have added Guided Backprop and Input×Gradient to our analysis. Results are consistent: architecture rankings are strongly correlated (ρ≈0.91), and patterns remain unchanged (CNNs biased; ViTs/MLPs neutral).
>   Benchmark behavior is also consistent: under corner-shift training, CSI shifts toward the scale-specific ideal (≈ 0.5) but not fully, and trained–random similarity patterns remain stable across architectures.
> - **CSI definition and negative values.** We thank the reviewer for this observation. The negative values arise because we use a signed displacement to capture the direction of saliency shift relative to the image center. We agree the definition should be clarified and will revise the text and caption accordingly. We retain the signed formulation, as it provides meaningful directional information in this benchmark.
> - **Pretrained vs fine-tuned models.** (reviewer-suggested comparison). We evaluated standard ImageNet-pretrained models (without Center/Corner fine-tuning) on both Center Shrunk and Corner Shrunk inputs, as proposed (see https://github.com/anonymousresearcher672/anonymous-project-2026/blob/main/fig5_ex.png). We note that accuracy drops substantially under these distribution shifts (e.g., ~70% → ~20% for ResNet18), which limits quantitative interpretation. In contrast, fine-tuned models achieve comparable validation accuracy across center- and corner-shifted settings (Table 1) and produce stronger, more consistent shifts toward the displaced object.
>   This shows the benchmark isolates architectural bias when the model functions correctly, and that the remaining center bias is not due to insufficient learning but reflects an interaction between learned behavior and architectural priors.
> - **CB-AUC definition.** (corner vs full band). While we use corner regions, conclusions do not depend on this choice. As an additional robustness check, we replaced the corner-only region with the full outer band surrounding the center. Results remain highly consistent (Spearman ≥ 0.986; Pearson ≈ 0.997), with nearly identical architecture rankings. This aligns with additional robustness analyses (Reviewer 1cJf: crop size; Reviewer vRuw: scale/position variations), showing that center-bias conclusions are stable across region definitions.
> - **Initialization.** We use standard torchvision/timm implementations (weights=None), which follow architecture-specific default initialization schemes (e.g., Kaiming for CNNs, truncated normal for transformer-based models). We do not vary initialization methods, as our goal is to study architectural effects under standard setups.
> - **Mechanistic evidence.** (variance to gradient). We have performed additional empirical analyses. Across both ResNet18 and ConvNeXt-Tiny, we observe: (1) higher central activation variance, (2) increasing center–periphery variance gap with depth, and (3) smooth radial decay. These patterns are consistent with the predicted relationship between activation variance and gradient magnitude and demonstrate that the mechanism persists in modern CNN architectures (see variance maps and radial profiles: https://github.com/anonymousresearcher672/anonymous-project-2026/blob/main/variance_ac_depth.png, https://github.com/anonymousresearcher672/anonymous-project-2026/blob/main/radial_profile.png).
> - **ImageNet object location statistics.** To support the assumption of object-centered data, we quantified object locations using the official bounding-box annotations. Defining the central 50% × 50% region, we find that 76.8% of object centers lie within this region, and 92.5% of images contain at least one centrally located object. This confirms strong dataset-level center bias, motivating our benchmark.
> - **Reproducibility.** We will release the full codebase upon acceptance.

---

> > ### Author Rebuttal · Reviewer_9TdE · 2026-04-03
> >
> > I would like to thank the authors for the **supplementary experiments** provided in the rebuttal. They address a majority of my questions. I have a couple of remaining points to address:
> >
> > - In the rebuttal, the authors state: *''We focus on gradient-based saliency, where the mechanism relates to gradient propagation and activation variance. CAM or perturbation methods rely on different mechanisms and are outside our scope.''*
> >   I am not convinced by this distinction for CAM methods. For example, Grad-CAM directly relies on the gradient of the output logit with respect to the activation maps of a given layer. Could the authors clarify more precisely why Grad-CAM would not fall within their setting (or why to not include it)?
> >
> > - Regarding the negative CSI case, could the authors be more precise? I understand the intuition behind their explanation, but it would be helpful if they could provide the corrected CSI expression using the notation of the paper.
> >
> > - I would also like to reiterate the following question: is Figure 4 based on a single random initialization for each model?

---

> > > ### Author Response · Authors · 2026-04-05
> > >
> > > We thank the reviewer for the thoughtful follow-up questions and for noting that our additional experiments addressed the majority of the concerns. We appreciate this feedback and are happy to clarify the remaining points below.
> > > - **Grad-CAM.** We agree with the reviewer that Grad-CAM is gradient-based in the sense that it uses gradients of the output logit with respect to feature maps. Our previous wording that CAM methods are “outside our scope” was imprecise; our intended distinction is that our mechanism specifically analyzes gradient propagation to the input, whereas Grad-CAM operates on intermediate feature maps and depends on layer selection.
> > >
> > >   To address this directly, we have included Grad-CAM analysis using a consistent late-convolution layer across CNN architectures, as suggested by the original paper. Under this setup, most CNNs exhibit positive center bias, broadly consistent with our main finding, although the effect is generally weaker compared to input-gradient methods (vanilla, Integrated Gradients, SmoothGrad, Guided Backprop, and Input×Gradient). We also observe greater variability across random seeds (standard deviation up to ~0.17 for some models) and dependence on target-layer choice.
> > >
> > >   We will include these results and clarify this distinction and setup in the revision.
> > > | Architecture    | Mean Center Bias | Std (across seeds) |
> > > |-----------------|------------------|--------------------|
> > > | ResNet18        | 0.81             | 0.17               |
> > > | VGG11           | 0.78             | 0.01               |
> > > | DenseNet121     | 0.77             | 0.17               |
> > > | AlexNet         | 0.72             | 0.11               |
> > > | MobileNet_V2    | 0.65             | 0.03               |
> > > | SqueezeNet1_0   | 0.64             | 0.06               |
> > > | EfficientNet_B0 | 0.64             | 0.04               |
> > > | ShuffleNet_V2   | 0.58             | 0.08               |
> > > | InceptionV4     | 0.50             | 0.12               |
> > > - **Signed CSI definition** We thank the reviewer for the request for clarification. We previously described the intuition for negative values; here we make the definition explicit.
> > >
> > >   In the corner-shift benchmark, we use a **signed** version of CSI so that the metric captures both the magnitude and direction of saliency displacement relative to the image center.
> > >
> > >   Using the notation of the paper, let (x_com, y_com) denote the saliency center of mass and let c = (c_x, c_y) denote the image center. For the top-left corner setting, we define:
> > >
> > >   CSI(x) = [(c_x - x_com) + (c_y - y_com)] / (c_x + c_y)
> > >
> > >   Under this formulation:
> > >
> > >   - CSI(x) ≈ 0 indicates saliency near the image center,
> > >   - CSI(x) > 0 indicates displacement toward the target top-left corner (with ≈ 0.502 as the ideal value),
> > >   - CSI(x) < 0 indicates displacement in the opposite direction.
> > >
> > >   We will clarify this in the text and captions to explicitly state the signed formulation and avoid ambiguity.
> > >
> > > - **Figure 4 (random initialization)** Yes, Figure 4 in the submission is based on a single random initialization per architecture; the ridgeline variability in Fig. 4 reflects variation across images, not across seeds. To address this directly, we repeated the random-initialization experiment over 5 seeds for vanilla saliency. The architecture-level center-bias ordering remains highly stable across seeds, with small seed-to-seed variation relative to the between-architecture differences. We will clarify in the revision that the original figure used one initialization, and add this seed-robustness analysis in the appendix. We thank the reviewer for highlighting this point, which helped strengthen the robustness of our analysis.
> > > | Architecture     | Mean CB-AUC | Std (across seeds) |
> > > |------------------|------------|--------------------|
> > > | InceptionV4      | 0.920      | 0.006              |
> > > | AlexNet          | 0.793      | 0.001              |
> > > | ResNet18         | 0.753      | 0.010              |
> > > | EfficientNet_B0  | 0.730      | 0.040              |
> > > | MobileNet_V2     | 0.715      | 0.007              |
> > > | SqueezeNet1_0    | 0.696      | 0.010              |
> > > | DenseNet121      | 0.671      | 0.011              |
> > > | VGG11            | 0.625      | 0.005              |
> > > | ShuffleNet_V2    | 0.580      | 0.017              |
> > > | MLP_Mixer_B16   | 0.520       | 0.014                |
> > > | ViT_L16_timm    | 0.514       | 0.002                |
> > > | ViT_B16_timm    | 0.513       | 0.001

---

### Official Review · Reviewer_1cJf · 2026-03-13

**Soundness:** 3
**Presentation:** 3
**Significance:** 3
**Originality:** 3
**Overall Recommendation:** 4
**Confidence:** 3

**Summary:**

The following paper provides deeper explanations of a well-known phenomenon of saliency maps for CNNs. It conducts various tests and provides explanations why the saliency maps of randomly initialized and trained CNNs look so similar. It proposes metrics to measure the center focused bias in models and is induced by zero padding and receptive field growth in CNNs. The paper, also, shows that this phenomenon is no longer present or as pronounced in non-CNN architectures such as  Vision Transformers. Additionally, the paper introduces corner-shift and center-shift benchmarks and showcases the training can shift the salency maps towards the shifted region.

**Compliance With Llm Reviewing Policy:**

Affirmed.

**Key Questions For Authors:**

On figure 7 we observe that some of the CNNs such as Inception of Resnet18 have much higher similarity between trained vs random compared to VGG16 for example. Why is that the case ?

**Limitations:**

The limitations of the work are not discussed

**Strengths And Weaknesses:**

**Pros**
+ The paper is very well written with a clear motivation, connection to the previous work in the field and very interesting observations about the connection between the architectural properties of a network and salency maps.
 + The paper also did thorough experiments with different types of model architectures, studying the effects both for padding and receptive fields.


**Cons**
+ The metric proposed in the paper depends on the crop size and it is unclear how the  results will change if we change the crop size. Also, the work seems to assume that there is one central object in the image but in the reality there could be more one central object.
+ There is also a well-known phenomenon that many saliency map method use input as a multiplier which results the saliency maps acting as edge detectors. It might be interesting to compare with works that attempt to quantify the fidelity of saliency maps.
+ Since the paper is only focusing on computer vision and does not explore other modalities, it might be a better fit for a computer vision conference.

---

> ### Author Rebuttal · Authors · 2026-03-31
>
> We appreciate the reviewer’s detailed and constructive feedback. Below we clarify key points and outline revisions:
> - **Crop size / benchmark robustness.** We will clarify the robustness of crop size and include additional analysis in the appendix. As shown below, center-bias rankings are highly stable across different crop sizes (Spearman ≈ 0.97–1.00), indicating that our conclusions are not sensitive to this choice. We also performed additional robustness checks on the benchmark (varying object scale and placement; see Reviewer vRuw, (Corner-shift benchmark as stress test + robustness)), which show consistent trends across settings.
> | Frac Size | Pearson Mean Corr | Spearman Rank Corr |
> |-----------|-------------------|--------------------|
> | 0.3       | 0.9952            | 0.993              |
> | 0.4       | 0.9979            | 0.993              |
> | 0.5       | 1.0000            | 1.000              |
> | 0.6       | 0.9968            | 1.000              |
> | 0.7       | 0.9818            | 0.972              |
>
>
> - **Multiple objects / object location.** Our analysis does not assume a single central object; rather, it depends on the population-level spatial distribution of objects. Using official ImageNet bounding-box annotations, we find that 92.5% of images contain at least one centrally located object, despite many images having multiple objects. To isolate this effect, our corner-shift benchmark deliberately breaks this alignment by controlling object location, allowing us to disentangle architectural bias (center) from dataset statistics.
> - **Edge-detector effects vs architectural bias.** We agree that edge sensitivity is an important factor (Adebayo et al., 2018). Our results complement this line of work by showing that even when edge structure is removed (pixel-shuffled inputs), a strong center-focused pattern persists (Fig. 1), indicating an additional architecture-induced spatial prior beyond edge effects. Regarding fidelity-based evaluation, our goal is orthogonal: rather than assessing whether saliency reflects model predictions, we analyze systematic spatial structure induced by architecture. We will clarify this distinction and add discussion of fidelity-based metrics in the revision.
>
> - **Why similarity differs across architectures (Fig. 7).** We agree this is an important point. Empirically, architectures such as Inception and ResNet exhibit stronger center bias (Fig. 4), and these same architectures show higher trained–random similarity. When this center bias is reduced (e.g., under corner-shifted inputs), the similarity correspondingly decreases, suggesting that center bias contributes to the observed similarity.
>     From an architectural perspective, these models are deeper and involve more aggressive receptive field growth and padding effects, which may amplify center–periphery differences compared to simpler, sequential architectures such as VGG.
> - **Scope.** While our experiments focus on CNN architectures, the core contribution is conceptual—identifying architecture-induced bias in interpretability evaluation, which is relevant beyond specific model families.
> - **Limitations.** We have noted the reviewer’s comment and will expand the discussion to highlight key limitations.

---

> > ### Author Rebuttal · Reviewer_1cJf · 2026-04-05
> >
> > Thank you the authors for taking time and addressing my comments and questions.
> >
> > 1. Regarding Figure 7: Thank you for the explanations. I think I'm still not clear why for VGG11 the similarity score is negative but for Alexnet for example it is positive. Also, it is unclear why the gap between center and corner is large for some networks   and for some cases, e.g. ElasticNet_B0 the gap is almost not there. It looks like for ElasticNet_B0 the corner similarity is even larger ?

---

> > > ### Author Response · Authors · 2026-04-07
> > >
> > > We thank the reviewer for this helpful follow-up. Figure 7 is intended to show that architectural center bias acts as a **confounding factor** in similarity-based evaluation, while architectural-specific saliency patterns (see Appendix Fig. 15) still play a role.
> > >
> > > For architectures with strong center bias (e.g., Inception-V4, AlexNet), both random and trained saliency maps concentrate near the image center under standard settings, leading to higher similarity for centered inputs and a clear gap between center and corner conditions, consistent with our proposed mechanism (see Fig. 6). In these cases, center bias dominates the similarity behavior.
> > >
> > > In contrast, for architectures with weaker center bias (e.g., VGG11, MobileNet, SqueezeNet), architectural-specific saliency patterns play a larger role. While trained models consistently focus on the shrunken images, random models often exhibit more diffuse or background-dominated patterns. Appendix Fig. 15 provides qualitative examples illustrating such architecture-specific patterns across models.
> > >
> > > For example, in VGG11 (see Appendix Fig. 15), random saliency maps frequently assign relatively high magnitude to shuffled background regions, sometimes exceeding that of natural image regions. This can lead to negative similarity values, as similarity is driven by global pattern correlation: random models emphasize background regions, while trained models suppress them. We also observe other architecture-specific behaviors (e.g., EfficientNet random models exhibit patch-like patterns and do not clearly highlight object edges), though their precise relationship to similarity varies.
> > >
> > > When center bias is weak, these architectural effects become more influential, and similarity is driven more by global differences between object-focused and diffuse patterns. This can reduce the center–corner gap or even lead to negative similarity under correlation-based metrics.
> > >
> > > We also note that the bars in Fig. 7 are center-aligned (not end-aligned); we believe that the case referred to by the reviewer corresponds to MobileNet (rather than EfficientNet-B0), which has  corner similarity comparable to or larger than center similarity.
> > >
> > > **Overall, Fig. 7 shows that center bias dominates similarity when strong, while architecture-specific saliency patterns become more prominent when center bias is weak.**

---

### Decision · Program_Chairs · 2026-04-30

**Decision:**

Accept (regular)

**Comment:**

This article identifies center bias (arising from convolutional padding) as an important confounder in saliency maps. While the components (center bias on one side, and broader issues with saliency maps on the other side) were established previously, an in-depth investigation of their intersection is described as a useful and novel contribution by reviewers. During rebuttal time, the authors provided helpful clarifications and additional experiments. In particular, they confirm that their findings should be interpreted as center bias being a _confounder_, not the sole explanation. While reviewer vRuw perceives this as reducing the significance of the article, other reviewers (e.g. 9TdE) believe that this nonetheless provides a useful contribution to the field. On balance, the AC believes that the paper's observations are interesting enough to warrant publication at ICML, provided the authors incorporate the changes & clarified positioning regarding sole cause vs. confounder they promised during the rebuttal.